# Structural insights into the DNA topoisomerase II of the African swine fever virus

Jingyuan Cong[1,2,5], Yuhui Xin[1,2,5], Huiling Kang[3], Yunge Yang[1,2], Chenlong Wang[3], Dongming Zhao [4], Xuemei Li[1] ✉, Zihe Rao[1,3] ✉ & Yutao Chen[1] ✉

Type II topoisomerases are ubiquitous enzymes that play a pivotal role in modulating the topological configuration of double-stranded DNA. These topoisomerases are required for DNA metabolism and have been extensively studied in both prokaryotic and eukaryotic organisms. However, our understanding of virus-encoded type II topoisomerases remains limited. One intriguing example is the African swine fever virus, which stands as the sole mammalian-infecting virus encoding a type II topoisomerase. In this work, we use several approaches including cryo-EM, X-ray crystallography, and biochemical assays to investigate the structure and function of the African swine fever virus type II topoisomerase, pP1192R. We determine the structures of pP1192R in different conformational states and confirm its enzymatic activity in vitro. Collectively, our results illustrate the basic mechanisms of viral type II topoisomerases, increasing our understanding of these enzymes and presenting a potential avenue for intervention strategies to mitigate the impact of the African swine fever virus.

Topoisomerase II (Topo II) represents conserved enzymes pivotal in regulating the topological conformation of DNA molecules during a range of essential physiological processes, encompassing DNA packaging, replication, transcription, recombination and DNA repair[1,2]. These evolutionary conserved enzymes have been identified in eukaryotes, prokaryotes and viruses. There are two broad classes of type II topoisomerases: type IIA topoisomerases, which include DNA gyrase, topo IV and eukaryotic/viral topo II; and type IIB topoisomerases, which include Topo VI from plants, algae, archaea and some bacteria as well as homologues of eukaryotes Spo11[3,4]. The global architecture and substrates of type IIA topoisomerases are distinct from type IIB, but the catalytic modules and the mechanism are similar[5–7]. Regarding viruses, it is less known for viral Topo II but some evidence suggests that those Topo II are very similar to their respective

hosts[8]. Prior extensive structural and biochemical studies have elucidated key aspects of Topo II catalytic mechanisms[5]. These structures have unveiled the inherent architecture of Topo II, which is characterized by two principal heart-shaped functional domains: the ATPase domain, responsible for ATP hydrolysis to fuel the molecular machinery, and the central domain, which interacts with the DNA duplex and contains a catalytic tyrosine residue that cleaves the DNA duplex[2]. Facilitated by the DNA topological feature, such as a DNA supercoil, Topo II selectively associates with and induces a transient break in one DNA duplex, designated as the "G-segment". This orchestrated reaction generates a gap through which a second DNA duplex, known as the "T-segment," can translocate. Subsequently, the DNA break is resealed, restoring the integrity of the DNA structure. The complete process involves: G-segment breakage, T-segment

---

[1]National Laboratory of Biomacromolecules, Institute of Biophysics, Chinese Academy of Sciences, Beijing, China. [2]University of Chinese Academy of Sciences, Beijing, China. [3]Laboratory of Structural Biology, School of Medicine, Tsinghua University, Beijing, China. [4]State Key Laboratory for Animal Disease Control and Prevention, National African Swine Fever Para-reference Laboratory, National High Containment Facilities for Animal Diseases Control and Prevention, Harbin Veterinary Research Institute, Chinese Academy of Agricultural Sciences, Harbin, China. [5]These authors contributed equally: Jingyuan Cong, Yuhui Xin. ✉e-mail: lixm@ibp.ac.cn; Raozh@ibp.ac.cn; chenyutao@ibp.ac.cn

transportation, G-segment resealing, and release[2,9]. As a result, Topo II not only relaxes DNA supercoiling but also disentangles knots and catenates potentially detrimental to cellular function if left unresolved[1,10].

Despite the considerable elucidation of the structures of eukaryotic and prokaryotic Topo II enzymes, the structure and mechanism of full-length virus-encoded Topo II remains unclear[11,12]. In this study, we explored the characterization of a viral Topo II encoded by African swine fever virus (ASFV), which is the only member of the *Asfarviridae* family and the only known DNA arbovirus, and also the exclusive mammalian-infecting virus known to encode a functional Topo II[11,13–15].

ASFV stands as the causative pathogen of African swine fever (ASF), an exceptionally virulent and fatal hemorrhagic disease in swine since its first identification in Kenya in 1921[16]. Subsequently, the virus disseminated throughout Sub-Saharan Africa, Europe, South America and the Caribbean[17]. In 2018, it was infiltrated into the world's largest pig producer, China, thereby engendering a profound menace to the worldwide swine industry because of no clinically available drugs or vaccines[16,17]. ASFV contains a large DNA genome of 170-190 kb and encodes approximately 160 ORFs[18], lending its genome to the risk of intricate entanglements or knots, which necessitates the involvement of Topo II for resolution. The ORF P1192R encodes a functional Topo II (pP1192R) which is highly conserved across diverse ASFV isolates[14,19]. And P1192R knockdown assay by siRNA has been observed to impede viral infection by inhibiting gene transcription, highlighting pP1192R as a promising target for the development of drug and vaccines to counteract ASFV[13,18,20].

Here, we determined the first fully intact structure of a virus-encoded Topo II, including cryo-EM structures of the entire pP1192R-dsDNA complex in different conformations, alongside crystal structures of ATPase domain complexed with AMPPNP and ADP respectively. These structures clearly demonstrated that pP1192R had a high structure similarity with eukaryotic Topo II in spite of their low similarity on amino acid sequence, allowing analysis of the evolutionary trajectory of both the protein and the virus from a structural perspective[9,21]. Furthermore, our comparatively complete structures of intermediate catalytic states provide significant insights into the fundamental mechanisms of Topo II, which also hold promise in guiding strategies for the prevention and treatment of ASFV-related epidemics.

## Results

### Cryo-EM determination of pP1192R and its DNA duplex complex

To determine the structure of pP1192R, the full-length pP1192R and its central domain (408–1192) were expressed in *sf9* insect cells using the bac-to-bac expression system. The ATPase domain of pP1192R (1-434) was expressed in *E. coli* BL21(DE3). The enzymatic activities of purified protein samples were confirmed (Fig. 1a, b and Supplementary Fig. 1a, b). The apo pP1192R (full-length and the central domain) were plunged-frozen on an amorphous alloy film (R1.2/1.3, Au, 300 mesh) at the concentration of 0.8 mg/ml. In parallel, the full-length pP1192R was mixed with 52 bp DNA oligonucleotides, $Mg^{2+}$ and AMPPNP, which was subsequently plunged-frozen on an amorphous alloy film (R1.2/1.3, Au, 300 mesh) at the same concentration. The ATPase domain of pP1192R was mixed with AMPPNP/ADP, and concentrated to 5–15 mg/ml for crystallization.

The cryo-EM data were collected with a 300 kV FEI Titan Krios transmission electron microscope with a GIF-Quantum energy filter (Gatan) and K2-summit detector (Supplementary Fig. 1c, d). After motion correction, CTF estimation, 2D classification, 3D classification and 3D refinement, a total of 7 structures were obtained: the central domain of apo pP1192R in three distinct conformational states (Coil-open, Close and WHD-open) at resolution of 3.3, 3.4 and 4.3 Å respectively; the DNA-bound pP1192R complex in four states: a structure encompassing the central domain (pP1192R_{CD-DNA}) at 3.2 Å resolution, along with three full-length pP1192R structures (pP1192R_{FL-3}) exhibiting

varying orientations of the ATPase domain, resolved at 5.6, 4.8 and 5.9 Å respectively (Fig. 1c and Supplementary Figs. 2–7). Statistics for cryo-EM data collection is summarized in Supplementary Table 1 and Table 2. After model building and refinement, the final structures were validated using program CCP4[22] and PHENIX[23]. All information about these structures were provided in Supplementary Table 4.

### Crystal structures of ATPase domain with AMPPNP/ADP

Owing to the modest resolution of the ATPase domain in the cryo-EM density map, truncated pP1192R ATPase domain (1-434) with either AMPPNP or ADP was crystallized. The crystal structures of the ATPase domain complexed with AMPPNP and ADP yielded high resolutions of 2.3 Å and 2.6 Å, respectively (Fig. 2c and Supplementary Fig. 7b, c). Both structures contain a protomer within the crystallographic asymmetric unit and these high-resolution crystal structures were employed in the cryo-EM structure construction. The statistics of the diffraction data and structural refinement are summarized in Supplementary Table 3. These two structures manifest minimal discernible conformational alterations, except notable interaction between E60 and K368 with the third phosphate of AMPPNP (Fig. 2c, d).

### Overall structure of pP1192R-DNA complex

As illustrated in Fig. 2, the atomic model of pP1192R-DNA complex encompasses a homodimer of the GHKL, Transducer, TOPRIM, WHD, Tower and Coiled-Coil subdomains, alongside a curving DNA duplex. Within this sophisticated assembly, the GHKL and Transducer subdomains constitute the ATPase domain. The TOPRIM, WHD, Tower and Coiled-Coil subdomains collectively form the central domain (or DNA-binding/cleavage domain). A pliable short linker (aa 410-420) interconnects these two domains, accounting for the wobbling of the ATPase domain. The TOPRIM subdomain is characterized by conserved acidic residues (E438, D539 and D541) that collaborate to form a conserved binding site for $Mg^{2+}$, which is found in other enzymes including type IA topoisomerases, DnaG-type primases, OLD family nucleases and RecR proteins[24]. The WHD subdomain contains a conserved Tyrosine (Y800) residue, pivotal for nucleophilic attack, along with a conserved Arginine (R799) crucial for stabilizing the cleavage intermediate. Collaboratively, the TOPRIM, WHD and Tower subdomains constitute a positively charged DNA binding groove that accommodates a DNA duplex (Supplementary Fig. 8a). Situated at bottom, the Coiled-Coil subdomains form the C-gate, and their dedimerization orchestrates the release of the T-segment.

### Structural insights into the interactions between pP1192R and DNA duplex

As depicted in Fig. 3, the DNA duplex engages in an extensive interaction with the DNA binding groove of pP1192R. The interface area defining the interaction between pP1192R and DNA spans about 4100 Å² (Supplementary Fig. 8a). This intricate interaction can be categorized into two distinct regions: (1) Within the central domain (Fig. 3a), the WHD, Tower and TOPRIM subdomains collaborate to envelop and interact with the DNA. The WHD subdomain is characterized by the presence of three conserved helices H1 (716-728), H2 (738-748) and H3 (755-766) (Fig. 3b), alongside a crucial β-hairpin motif, denoted as wing-1 (778-805), which contains the catalytic residue Y800 (Supplementary Fig. 8c). Specifically, H3 effectively inserts into the major groove of the DNA, while H1 and H2 act as protective barriers, limiting the depth of H3 intrusion into this groove. Each recognition helix exclusively contacts with the backbone of a single DNA strand. The catalytic Y800 on wing-1 comes into close proximity with the DNA backbone, enabling its role in the phosphodiester bond breakage (Supplementary Fig. 8c). Meanwhile, the Tower subdomain presents an additional β-hairpin, referred to as wing-2 (847-860). This wing-2 motif adeptly intercalates itself amidst the base stacks of the bound DNA, facilitating a global DNA bending of ~150° and a localized

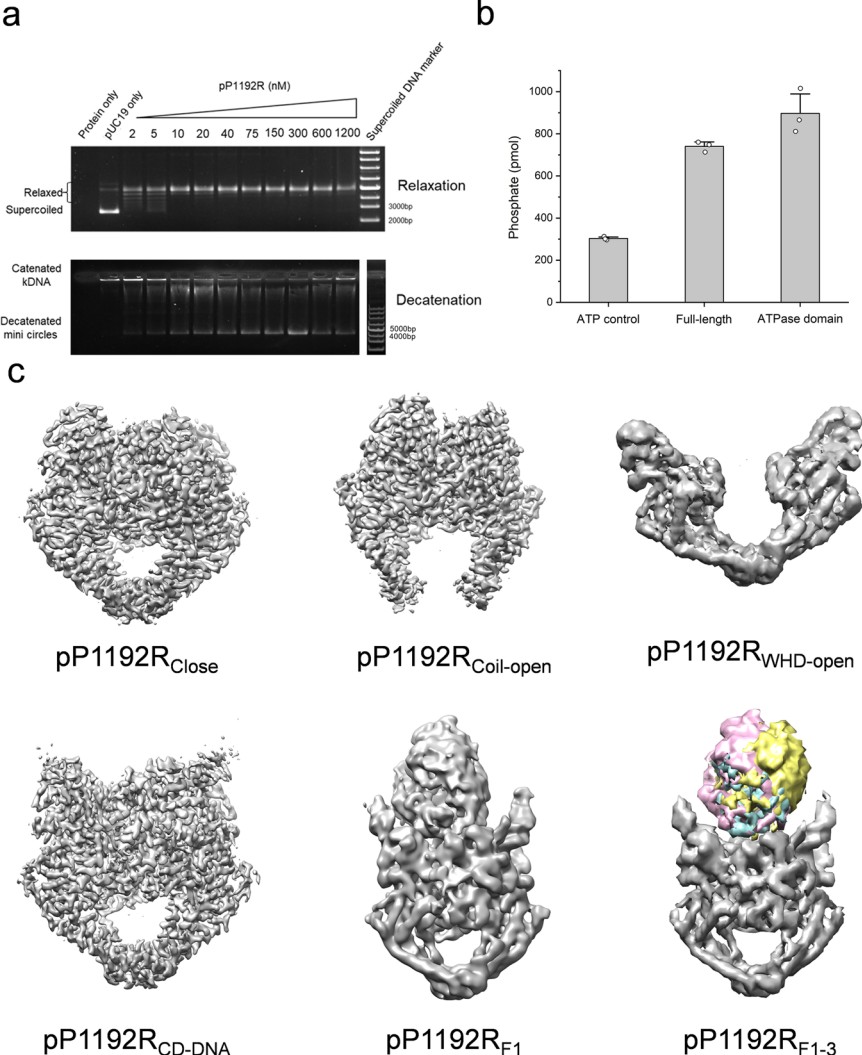

**Fig. 1 | Biochemical assays and cryo-EM maps of pP1192R and complexes. a** In vitro relaxation and decatenation activity of the purified pP1192R. Source data are provided as a Source Data file. **b** In vitro ATPase activity of the purified full-length pP1192R and truncated ATPase domain. The y-axis represents the total amount of free phosphate ions measured in the solution over 30 minutes. Data are presented as mean values +/− SEM for 3 independent replicates ($n = 3$). Individual data points are also plotted. The source data are provided as a Source Data file. **c** The final cryo-EM density maps of pP1192R and pP1192R-DNA complexes in distinct conformations (Colored regions represent comparison of the ATPase domain with different orientations: pP1192R$_{F1}$ (pink), pP1192R$_{F2}$ (yellow) and pP1192R$_{F3}$ (blue)).

DNA conformational transition from the B-form to the A-form (Fig. 3a, c). Notably, the intercalating residue on wing-2 of pP1192R is uniquely substituted by proline, in contrast to the universally conserved isoleucine among all other homologous type IIA topoisomerases[25], underscoring its distinct evolutionary pathway. Moreover, the concerted participation of S953, R956, R1004, H1010, and H1012 of the Tower domain serves as a supportive framework for the outermost ends of the bound DNA (Supplementary Fig. 8b). In addition to the Mg$^{2+}$-binding residues (E438, D539 and D541) within the TOPRIM subdomain, a set of other amino acid residues (M475, N476, K480 and K547) are also observed to engage in direct interactions with the DNA backbone (Fig. 3d). (2) Within the ATPase domain, a subtle yet discernible interaction was observed between the terminus of the bent DNA duplex and the periphery of the Transducer subdomain of one protomer (Fig. 3e). This unique interaction arises due to the distinct tilt observed in the ATPase domain, presenting a departure from the previously reported K-loop-DNA interaction[2].

### Analysis of the pP1192R conformations

During the course of cryo-EM image processing from the dataset, we determined three distinct full-length pP1192R structures, each

characterized by a unique orientation of the ATPase domain in relation to the central domain: pP1192R$_{F1}$, pP1192R$_{F2}$, and pP1192R$_{F3}$ (Fig. 1c and Supplementary Fig. 9a–c). In pP1192R$_{F1}$, the ATPase domain adopts an approximately 115° orthogonal orientation above the central domain, featuring a discernible tilt towards the DNA terminal (Supplementary Fig. 10a). In contrast, the pP1192R$_{F2}$ structure has the ATPase domain positioned at an approximately 95° orthogonal orientation above the central domain, extending almost vertically upward. Meanwhile, within the pP1192R$_{F3}$ structure, the orientation angle remains consistent with pP1192R$_{F1}$ at around 120°, yet the ATPase domain distinctly tilts towards the TOPRIM subdomain as highlighted in Supplementary Fig. 10b. A comparative analysis reveals that, relative to the pP1192R$_{F1}$ and pP1192R$_{F3}$, the ATPase domain of pP1192R$_{F2}$ undergoes a significant translocation, moving a distance of ~30 Å (Fig. 4a, b and Supplementary Fig. 9b, c).

Regarding the central domain, the conformation of pP1192R$_{CD-DNA}$ resembles that of pP1192R$_{Close}$, with the exception that the TOPRIM and Tower subdomains within a protomer approach by approximately 7 Å. Similarly, the TOPRIM and Tower subdomains between the two protomers diverge by approximately 4 Å, generating a notable cavity to accommodate a T-segment positioned atop the G-segment DNA, in

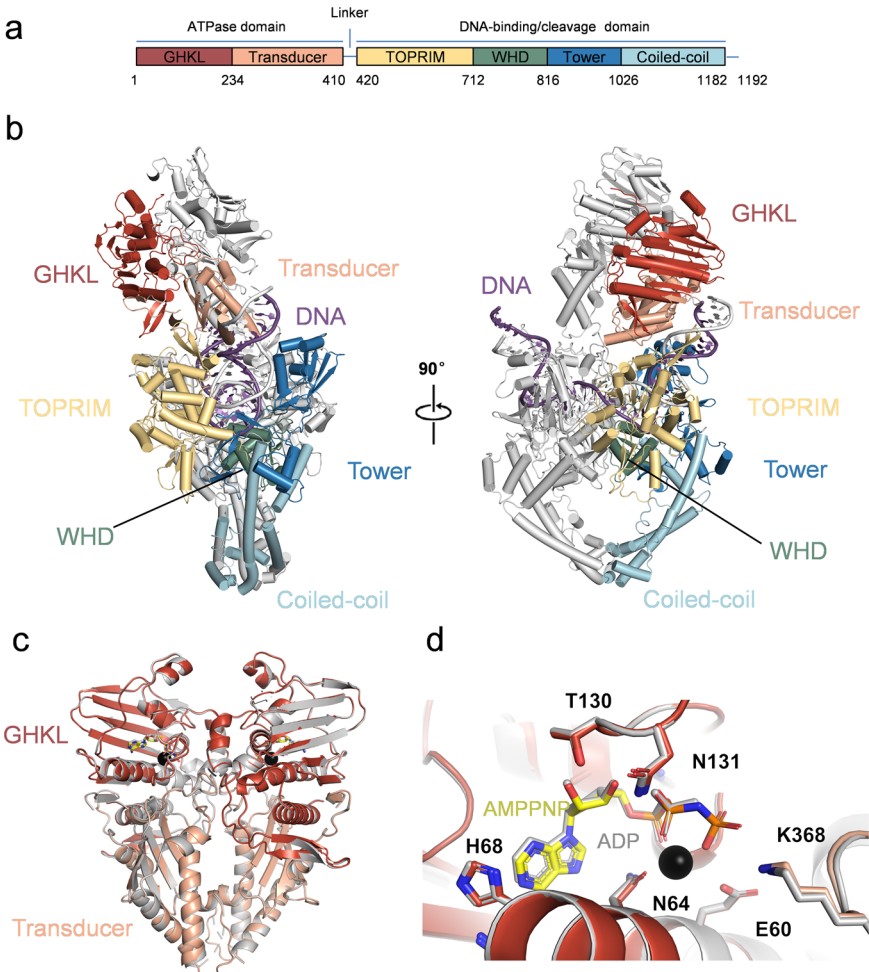

**Fig. 2 | Overall structure of pP1192R$_{\text{F1-DNA}}$ complex. a** Schematic diagram of the domain architecture of pP1192R, including six domains: GHKL (red), Transducer (orange), TOPRIM (yellow), WHD (green), Tower (blue) and Coiled-coil (cyan). **b** Cartoon representation of pP1192R$_{\text{F1-DNA}}$ complex. One protomer is colored by domains with coloring scheme identical to that in (**a**), the other is shaded gray. DNA is colored by purple. The unresolved linker regions are connected by dashed lines. The right structure is rotated 90° along the vertical axis. **c** Superimposition of crystal structures of ATPase domain complexed with AMPPNP (colored) and ADP (gray), respectively. **d** Close-up view of ATPase domain catalytic center (key residues are shown as sticks and Mg$^{2+}$ is represented as sphere).

agreement with prior reports[9] (Fig. 4c and Supplementary Fig. 10c). A comparative analysis of the three apo central domain structures (pP1192R$_{\text{Close}}$, pP1192R$_{\text{Coil-open}}$ and pP1192R$_{\text{WHD-open}}$) reveals that notable conformational shifts are evident within these structures. In comparison with pP1192R$_{\text{Close}}$, the pP1192R$_{\text{Coil-open}}$ structure exhibits a conspicuous -17 Å gap, resulting from the separation of two Coiled-Coil subdomains (Fig. 4d). In pP1192R$_{\text{WHD-open}}$, the interface of WHD-WHD dimer undergoes a separation of approximately 26 Å, while the Tower subdomains expand outward by -27 Å (Fig. 4e), similar to the *S. cerevisiae* Topo II structure[21] (pdb:1BGW). These prominent conformational changes demonstrate distinct states associating with the cleavage of the G-segment and the transport of the T-segment by pP1192R (Supplementary Fig. 10d).

### Conserved features of the pP1192R catalytic center

The catalytic center of pP1192R is composed of both the Mg$^{2+}$-binding region within the TOPRIM subdomain and the catalytic residue Y800 located in the WHD subdomain. Specifically, acidic residue E438, D539 and D541 of the TOPRIM subdomain engage with the Mg$^{2+}$. Serving as an intermediary bridging ligand, Mg$^{2+}$ further establishes electrostatic interactions with the DNA backbone phosphate (Fig. 5a, b). The positioning of Y800, which is responsible for the nucleophilic attack, varies greatly among different conformations. In pP1192R$_{\text{CD-DNA}}$, the distance

between Y800 and the nearest DNA backbone phosphate is 3.3 Å, and the distance between Y800 in two protomers amounts to 27.4 Å (Fig. 5b and Supplementary Fig. 11a). In both pP1192R$_{\text{Coil-open}}$ and pP1192R$_{\text{Close}}$ state, this distance between Y800 in two protomers expands to approximately 34.2 Å (Supplementary Fig. 11b). Upon modeling the identical DNA from the pP1192R$_{\text{CD-DNA}}$ structure into the corresponding positions of the above two structures (pP1192R$_{\text{Coil-open}}$ and pP1192R$_{\text{Close}}$), the distance between Y800 and the DNA backbone phosphate increases to 7.4 Å and 6.4 Å respectively, distances inadequate for nucleophilic attack (Fig. 5c). In contrast, binding of DNA to the central domain triggers repositioning of Y800, resulting in an approximately 5.1 Å approach to the phosphate, preparing for the nucleophilic attack (Fig. 5d). Compared with pP1192R$_{\text{Coil-open}}$ and pP1192R$_{\text{Close}}$, pP1192R$_{\text{WHD-open}}$ exhibits an apparent separation of -26 Å between the WHD subdomains (Fig. 4e). Consequently, this structural alteration leads to a pronounced angular change of -40° in the orientation of the two catalytic sites of Y800, accompanied by minor alterations in distance (-2 Å) (Supplementary Fig. 11c). Compared with pP1192R$_{\text{CD-DNA}}$, two catalytic Y800 of pP1192R$_{\text{WHD-open}}$ also rotate by ~40° and the distance stretches to 32.6 Å (Supplementary Fig. 11d). The diverse positions of Y800 across these conformations serve as indicators of the distinct functional states adopted by pP1192R.

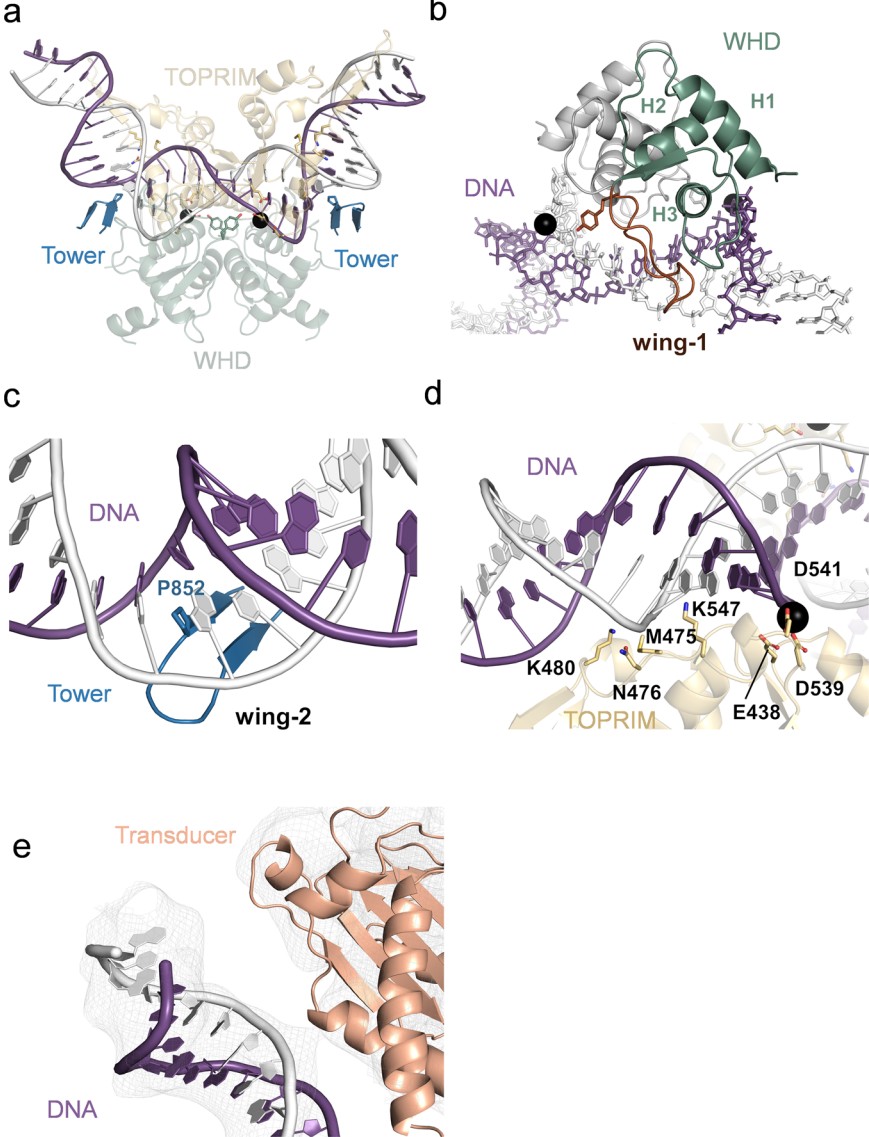

**Fig. 3 | Interactions between G-segment DNA and pP1192R subdomains. a** the interface between the central domain and DNA. **b** WHD-DNA interaction, Wing-1 is colored in brown. **c** Tower-DNA interaction, the DNA-binding β-hairpin (wing-2) is highlighted. **d** TOPRIM-DNA interaction. **e** Close-up view of the transducer subdomain's interaction with DNA backbone. (DNA and protein are shown in cartoon and colored as Fig. 2. Black sphere represents $Mg^{2+}$ and key residues are shown as sticks and labelled).

### In vitro interaction of pP1192R with DNA crossovers

Previous studies indicated a tendency of type II DNA topoisomerases to preferentially bind to DNA crossovers or juxtapositions[26]. To explore whether pP1192R also displays this preference, atomic force microscope (AFM) was used to visualize the interaction between pP1192R and supercoiled plasmid DNA. As shown in Supplementary Fig. 12, pP1192R displayed an inclination to bind to DNA crossover regions, similar to other topoisomerases[27,28]. Subsequently, an in vitro constructed four-way DNA was incubated with pP1192R. Results from Size Exclusion Chromatography (The ratio of absorbance values at 260 nm and 280 nm (-1.55) indicates a prominent nucleic acid binding compared with Supplementary Fig. 1a (-0.36)) and Electrophoretic mobility shift assay (EMSA) indeed exhibits the pronounced affinity of pP1192R for this particular DNA conformation (Fig. 6), consistent with previous findings[26].

### Discussion

Throughout the life cycle of ASFV, pP1192R plays a significant role in regulating the topological configurations of viral DNA molecules. In

this study, we determined the structures of pP1192R in various states and confirmed that the proteins we purified are functional through in vitro enzymatic assays, thereby effectively validating the correspondence between protein structure and function. Briefly, distinct conformational states within pP1192R exist: (1) A dimeric ATPase domain complexed with two substrates (AMPPNP/ADP) and its three distinct tilt angles atop the central domain. (2) Dynamic open/close conformations evident in the dimeric interface formed by the TOPRIM/WHD /Coiled-Coil subdomain. (3) Varied positions and orientations of the two catalytic Y800 residues. (4) The presence/absence of DNA duplex. In total, these findings offered a comprehensive elucidation of the molecular mechanism governing viral Topo II.

As indicated by our structural analyses, without DNA binding, the apo pP1192R undergoes dynamic conformational changes. Notably, three distinct apo conformations exist: $pP1192R_{Close}$, $pP1192R_{Coil-open}$ and $pP1192R_{WHD-open}$. Probably owing to higher stability, $pP1192R_{Coil-open}$ and $pP1192R_{Close}$ predominate among the extracted particles. During the review process of our manuscript, the states of $pP1192R_{Close}$ and $pP1192R_{Coil-open}$ were published[12]. We superposed the published

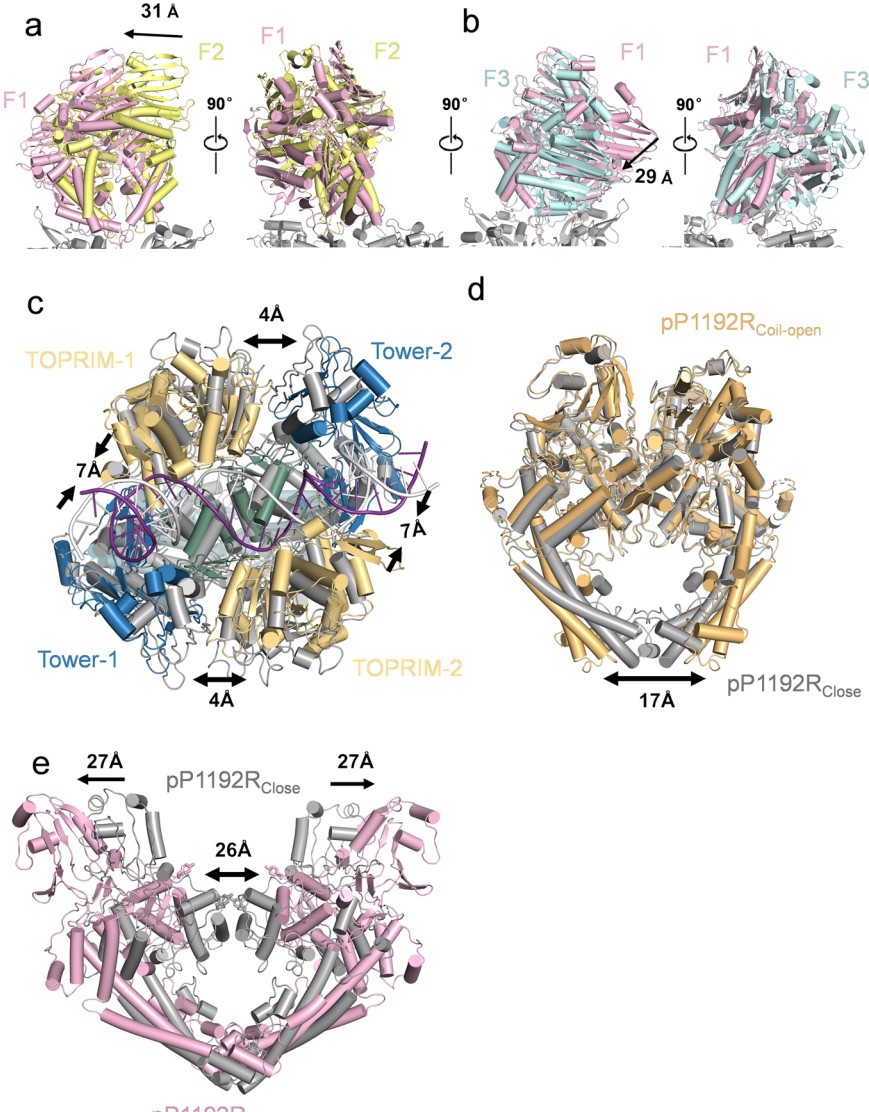

**Fig. 4 | Conformational changes of pP1192R and pP1192R-DNA complex.**
**a** Comparison of the ATPase domain between pP1192R$_{F1}$ (pink) and pP1192R$_{F2}$ (yellow). **b** Comparison of the ATPase domain between pP1192R$_{F1}$ and pP1192R$_{F3}$ (blue). **c** Comparison of conformational changes of the central domain between pP1192R$_{Close}$ (gray) and pP1192R$_{CD-DNA}$ (colored). **d** Superimposition of the central domain between pP1192R$_{Close}$ and pP1192R$_{Coil-open}$ (orange). **e** Comparison of conformational changes of the central domain between pP1192R$_{Close}$ and pP1192R$_{WHD-open}$ (pink). (The TOPRIM subdomain is omitted).

structures (PDB: 8J9Y, 8J9Z) with ours and confirmed these structures are essentially identical, with RMSD of 0.77 Å and 0.73 Å, respectively (Supplementary Fig. 13 a, b). Upon introduction of the DNA substrate and Mg$^{2+}$, the conformation of the central domain becomes more homogeneous. However, the ATPase domain retains a flexible conformation, which likely correlates with the energy transfer dynamics intrinsic to the DNA cleavage/stand passage process.

By comparing pP1192R structures with previous Topo II structures (pdb: 2RGR[25], 3L4J[29]), the structures are similar with RMSD of 2.53 Å and 2.05 Å, respectively (Supplementary Fig. 13 c, d). Focused on the catalytic tyrosine responsible for DNA cleavage, we proposed a more detailed catalytic mechanism, as depicted in Fig. 7. In the initial step (represented by pP1192R$_{Close}$ and pP1192R$_{Coil-open}$), the spatial separation between the catalytic tyrosine and the modeled DNA backbone phosphate measures 6.4 Å and 7.4 Å, respectively, exceeding the distance observed in 2RGR. This observation implies that pP1192R$_{Close}$ and pP1192R$_{Coil-open}$ are poised in a pre-binding state for the G-segment. In step 2 (represented by 2RGR), the Topo II forms a complex with the G-segment, while the catalytic tyrosine remains at a

distance of 5.5 Å from the DNA backbone phosphate. This distance is inadequate for nucleophilic attack, thereby signifying a pre-catalytic state. In step 3 (represented by pP1192R$_{CD-DNA}$), the conformation interconverts to an initial-cleavage state. Here, the proximity between the catalytic tyrosine and the DNA backbone phosphate contracts to 3.3 Å, an arrangement conducive to initiating DNA cleavage. Subsequently, in step 4 (represented by 3L4J), the formation of the 5'-phosphotyrosyl bonds is a hallmark of the cleavage state. Finally, in step 5 (represented by pP1192R$_{WHD-open}$), the G-segment undergoes further separation, facilitated by the dedimerization of WHD-WHD, thereby creating a gap primed for T-segment transport. In summary, our model demonstrates a progressive movement of the catalytic tyrosine toward the DNA backbone phosphate, resulting in the completion of the cleavage process.

In the full-length structures, a continuous wobbling characterizes the ATPase domain, suggesting a plausible correlation between the rotation of the ATPase domain and the oscillatory behavior of the central domain[9] (Fig. 4a, b). Notably, within the pP1192R$_{F1}$ structure, a segment of the transducer subdomain (proximate to S386) appears to

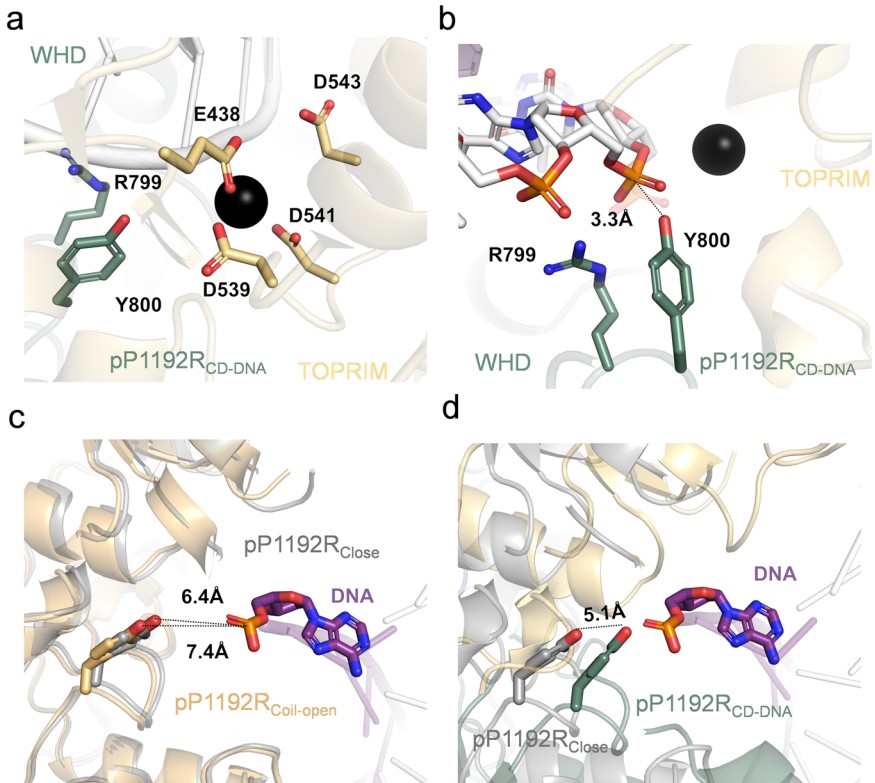

**Fig. 5 | Close-up view of pP1192R catalytic center. a**, **b** Close-up view of conserved catalytic residues interacting with DNA (R799 and Y800 on the WHD subdomain) and Mg²⁺ (E438, D539, D541, D543 on the TOPRIM subdomain). DNA and protein is shown in cartoon and colored as Fig. 2. Black sphere represents Mg²⁺ and the key residues are shown as sticks and labelled. **c** Close-up view of the interaction between catalytic Y800 and the DNA backbone phosphate in the superimposition of pP1192R$_{Close}$ and pP1192R$_{Coil-open}$ structures. **d** Close-up view of the interaction between catalytic Y800 and the DNA backbone phosphate in the superimposition of pP1192R$_{Close}$ and pP1192R$_{CD-DNA}$ structures. The Y800 undergoes a translocation of 5.1 Å.

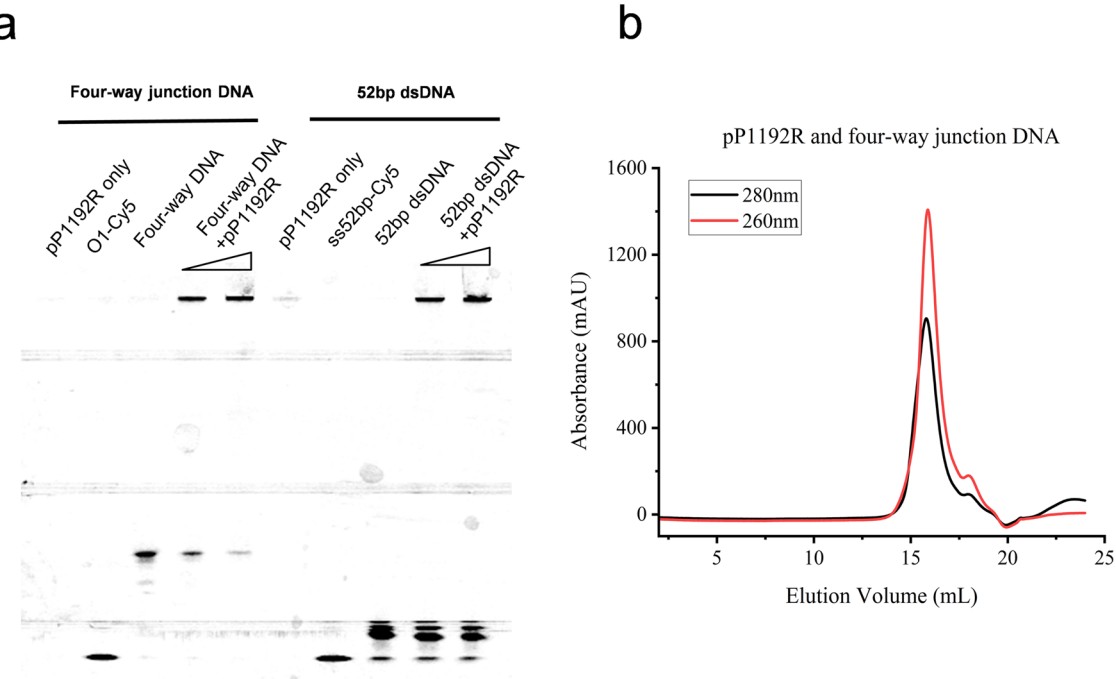

**Fig. 6 | Interaction of pP1192R with DNA crossovers. a** Result of EMSA. Four-way junction DNA and 52 bp dsDNA are assayed respectively. Protein-bound DNA shows slow migration. "O1-Cy5", "ss52bp-Cy5" indicate 5′-Cy5 fluorescently-labeled single strand DNA controls. The protein concentration is 1uM and 2uM, respectively. The concentration of labelled four-way DNA and 52bp dsDNA is 100 nM and 200 nM, respectively. EMSA experiments were reproduced three times independently. Source data are provided as a Source Data file. **b** Gel filtration of pP1192R and four-way junction DNA complex on a Superose 6 increase size-exclusion column. The ratio of absorbance value at 260 nm and 280 nm indicates a prominent nucleic acid binding.

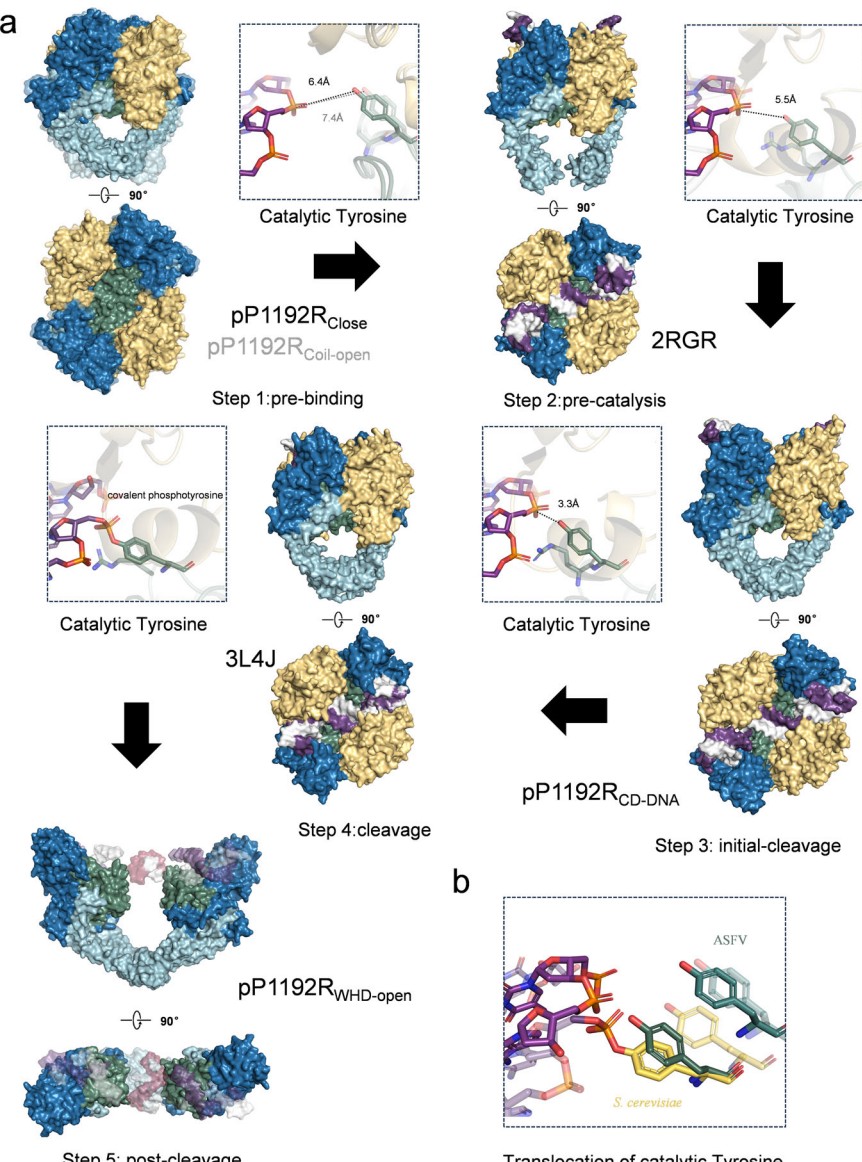

**Fig. 7 | A functional model of DNA duplex cleavage by Topo II elucidated through pP1192R. a** In step 1 (represented by pP1192R$_{Close}$ and pP1192R$_{Coil-open}$), the spatial separation between the catalytic tyrosine and the modeled DNA backbone phosphate measures 6.4 Å and 7.4 Å, respectively, exceeding the distance observed in 2RGR. This observation implies that in this state Topo II is poised in a **pre-binding state** for the G-segment. Progressing to step 2 (represented by 2RGR), the Topo II forms a complex with the G-segment, while the catalytic tyrosine remains at a distance of 5.5 Å from the DNA backbone phosphate. This distance proves inadequate for nucleophilic attack, thereby signifying a **pre-catalytic state**. Advancing to step 3 (represented by pP1192R$_{CD-DNA}$), the conformation is poised to interconvert into a **initial-cleavage state**. Here, the proximity between the catalytic

tyrosine and the DNA backbone phosphate contracts to 3.3 Å, an arrangement conducive to initiating DNA cleavage. Subsequently, step 4 (represented by 3L4J) witnesses the formation of the 5'-phosphotyrosyl bonds, a hallmark of the **cleavage state**. Finally, in step 5 (represented by pP1192R$_{WHD-open}$), the G-segment undergoes further separation, facilitated by the dedimerization of WHD-WHD, thereby creating a gap primed for T-segment transport termed **post-cleavage state**. DNA and protein is shown in surface and colored as Fig. 2. Close-up view of conserved catalytic residues interacting with DNA are shown as sticks with oxygen, nitrogen and Phosphorus atoms shown in red, blue and orange, respectively. **b** Close-up view of the translocations of catalytic tyrosine from ASFV (green) and *S. cerevisiae* (yellow).

engage with the G-segment (Fig. 3e), diverging from previously reported interactions involving the transducer subdomain and the G-segment[2]. The transducer subdomain orchestrates the coupling of DNA binding/cleavage with ATP hydrolysis and T-segment transport[5,9,30], but detailed mechanisms of the energy transfer from the ATPase domain to the central domain remain unsolved. The results of AFM show that the viral Topo II also prefers to bind the crossover site in a way similar to human Topo II[28]. The exact intricate mechanisms remain to be unveiled by higher-resolution full-length protein structures and complexes encompassing both T-segment and G-segment DNA. The AMPPNP in the structure of ATPase domain partially acts as

an inhibitor of pP1192R but lacks specificity. Previous study shows that Arctiin can block the ATP binding site[31], and our molecular docking confirmed its potential as a drug for pP1192R (Supplementary Fig. 14a). Furthermore, we conducted docking simulations with CT1[32], a Topo II inhibitor for trypanosomes, and found that Cyanotriazoles also have the potential to become inhibitors of pP1192R by affecting cleavage process (Supplementary Fig. 14b).

Considering the evolutionary perspective, ASFV encoding its own Topo II could be attributed to the fact that its large genomic DNA is prone to tangling. Besides, DNA replication of ASFV occurs in cytoplasmic viral factories, while the host's topoisomerase II mainly exists

in the nucleus[18]. By encoding its own Topo II, the virus may efficiently replicate within the host cell. It is worth noting that pP1192R has a low similarity on amino acid sequence with other species yet presents a high structure similarity with eukaryotic Topo II in human and yeast[9,12,21] (Supplementary Fig. 15). Structural comparison reveals that pP1192R embodies the most rudimentary form of the Topo II structure without additional C-terminal domain[5,14], suggesting two possibilities. Firstly, being evolutionarily more primitive, it is conceivable that pP1192R retained this elementary structure, representing an earlier stage in the evolutionary trajectory of Topo II. On the other hand, other proteins encoded by ASFV, such as pA104R, might engage in the interactions with pP1192R, thus taking on the corresponding regulatory function associated with the C-terminal domain such as the C-terminal of gyrase[33,34]. Certainly, definitive narrative regarding this evolutionary pathway necessitates further in-depth studies.

In conclusion, we obtained structures of full-length pP1192R in various states, bridging a critical gap in our knowledge concerning type IIA topoisomerases within the viral realm. These findings advance our comprehension of the mechanism underlying viral type IIA topoisomerases and offers structural insights into their evolutionary trajectories. Furthermore, the work holds the potential to make contributions to the prevention and treatment of African swine fever.

## Methods
### Protein expression and purification
The codon-optimized sequence of full-length P1192R (GenBank: MK128995.1) as well as the central domain (408-1192) was respectively inserted into a pFastBac1 expression plasmid (Invitrogen) with an 8× His-tag at the C-terminal, and expressed in *sf9* cells (Invitrogen: 11496-015) using a Bac-to-Bac expression system (Invitrogen) according to the manufacturer's instructions. The *sf9* cells were cultured with Sf-900II SFM. Following a 72 h post-infection (MOI = 2) at 27 °C, cells were harvested by centrifugation at 3000 g for 20 min. The cell pellet was resuspended using lysis buffer containing 50 mM Hepes (pH 7.2), 500 mM NaCl, 10% (v/v) glycerol, 2 mM Tris (2-carboxyethyl) phosphine (TCEP), and EDTA-free protease inhibitor. Subsequently, the resuspended cells were sonicated for 30 min, and the lysate was clarified by centrifugation at 16,000 *g* for 40 min at 4 °C. Clarified extract was loaded onto a Ni$^{2+}$-NTA column, then washed with lysis buffer plus 50 mM imidazole to remove unbound proteins and eluted with the lysis buffer plus 300 mM imidazole. The eluted sample was purified further over a Superose 6 Increase 10/300 GL column (GE Healthcare) in 50 mM Hepes, pH 7.2, 500 mM NaCl, and 2 mM TCEP, before concentrating and storing at −80 °C.

The DNA sequence encoding ATPase domain of pP1192R (1-434) was inserted into pET28a plasmid (Invitrogen) with 8× His tag at the C-terminal, and overexpressed in *E. coli* BL21(DE3) (Biomed), inducing with 0.5 mM Isopropyl β-D-1-Thiogalactopyranoside (IPTG) at 16°C overnight. The procedure of purification is the same as above except using lysis buffer with 50 mM Hepes, pH 7.2, 150 mM NaCl, 10%(v/v) glycerol 2 mM TCEP and EDTA-free protease inhibitor. The purified proteins were concentrated at 4 °C through centrifugation at 2,600 g using a centrifugal filter (Merck Millipore) with a molecular weight cutoff of 30 kD to a concentration of 5-15 mg/ml for crystallization.

### Nucleic acid preparation
A 52 bp DNA duplex was obtained by annealing two synthetic oligonucleotides[35]. Briefly, the oligonucleotides were dissolved in DNase-free water and each oligo was mixed at a 1:1 molar ratio, annealed by incubating at 95 °C for 2 min and then decreasing the temperature by 1 °C per minute until reaching 4 °C. A four-way junction DNA was obtained by annealing four synthetic oligonucleotides. The sequences were same as previous study[26] and the preparation procedure followed the aforementioned steps. For EMSA, a 5′-cy5 label was incorporated into one of the oligonucleotides (Oligo1) and one

strand of the 52 bp DNA. (synthesized by BGI, China). Detailed DNA sequences were summarized in Supplementary Table 5.

### BS3 cross-linked pP1192R-DNA complex
The purified full-length pP1192R was changed into cryo-EM buffer (50 mM Hepes (pH 7.2), 250 mM NaCl, 2 mM Mg$^{2+}$, and 2 mM TCEP) by mixing purified protein with a dilute buffer (50 mM Hepes (pH 7.2), 4 mM Mg$^{2+}$ and 2 mM TCEP) at 1:1 ratio. Then, it was mixed with a 52 bp dsDNA at a molar ratio of 1:1.2. The mixture was incubated for 10 min at room temperature. In order to stabilize the complex, AMPPNP was added to a final concentration of 0.5 mM. And freshly prepared BS3 was added to the fully reconstituted protein complex to a final concentration of 0.5 mM, incubating for 30 min at 4°C for cryo-EM sample preparation after centrifugation to remove aggregates.

### Cryo-EM sample preparation and data collection
The apo pP1192R and the BS3 (TargetMol) cross-linked pP1192R-DNA complex was diluted to 0.8 mg/ml in cryo-EM buffer. Before vitrification, the amorphous alloy film (R1.2/1.3, Au, 300 mesh) was glow-discharged 50 s at the condition of Hydrogen and oxygen. The grids were blotted for 3 s with -100% humidity at 4 °C and plunged into liquid ethane using a Vitrobot Mark IV (FEI). Cryo-EM micrographs were collected on a FEI Titan Krios transmission electron microscope operated at 300 kV with a GIF-Quantum energy filter (Gatan) and a Gatan K2-summit detector. All the Micrographs were automatically collected using Serial EM software (http://bio3d.colorado.edu/SerialEM/) in super-resolution counting mode with a pixel size of 0.65 Å (the nominal magnification of ×215,000) and a defocus range of −1.5 to −2.5 μm. The dose rate was 8 e⁻/pixel/s and exposure time of each image was 3.2 s to obtain an accumulative dose of ~60 e⁻/Å², fractioned into 32 frames. In total, 2964 images of the apo full-length pP1192R were collected; 4411 images of the apo central domain of pP1192R samples were collected; 2730 images at 0.65-pixel size and 5996 images at 0.82-pixel size (magnification of ×165,000) of the BS3 cross-linked pP1192R-DNA complex were collected.

### Image processing
RELION3.0.8[36] was used to process all images and reconstruct structures. 2964 micrographs of apo full-length pP1192R: The program MotionCorr2 with a 5 × 5 patch was used to correct the beam-induced motion and anisotropic magnification of the Micrographs. The program Gctf[37] was used to estimate the initial contrast transfer function (CTF). Micrographs with a resolution better than 4 Å were selected for subsequent data processing. About 10k particles from 40 micrographs picked by manual were subjected to 2D classification to generate templates for auto-picking against all the micrographs. A total of -1725 k particles were selected for reconstruction.

After 2D classification, -1725k particles was selected and further subjected to 3D classification. The selected particles were divided into five classes through 3D classification. The best class containing 150k particles was selected for 3D refinement and a density map of pP1192R$_{Coil-Open}$ with 3.3 Å was obtained. Moreover, a low-resolution density map of pP1192R$_{Close}$ was obtained.

4411 micrographs and 2730 micrographs were processed with procedures similar to those described above and density maps of pP1192R$_{Close}$ (3.4 Å) and pP1192R$_{CD-DNA}$ (3.2 Å) were obtained. 5996 micrographs were processed with procedures similar to those described above. Initially, a low-resolution density map of the full-length pP1192R was utilized as a reference to pick particles. These particles were subjected to 2D classification, 3D classification, and 3D refinement. Subsequently, focus 3D classification was used to achieve different tilts of the ATPase domain. Following further 3D refinement and post-processing, density maps of pP1192R$_{F1}$ (5.6 Å), pP1192R$_{F2}$ (4.8 Å) and pP1192R$_{F3}$ (5.9 Å) were obtained. Detailed procedures are included in Supplementary Figs. 2–6.

## Model building and refinement

Predicted model by AlphaFold2[38] was used for structure reconstruction of pP1192R. UCSF chimera[39] and COOT[40] were used for further optimization, and the real-space refinement was performed using PHENIX[23].

## Crystallization, diffraction data collection, and structure solution of ATPase domain

Initial crystal screening was performed by sitting drop method in 96-well plate using screening kits (Hampton research). 5–15 mg/mL truncated ATPase domain protein was incubated either with 1 mM AMPPNP or ADP (final concentration), then mixed with well solutions at volume ratio of 1:1 by mosquito. Best crystals were grown in hanging drop format by mixing 1 µL protein solution with 1 µL reservoir solution containing 21% PEG 3350, 100 mM Hepes (pH 7.2) and 200 mM $MgCl_2$ (or $Li_2SO_4$). Crystals were transferred for three minutes to the reservoir solution plus 15% glycerol before looped and flash frozen in liquid nitrogen.

Diffraction data were collected at the rotating-anode X-ray source MicroMax 007/Satun 944 HG/Varimax HF at a wavelength of 1.5418 Å (Institute of Biophysics, Chinese Academy of Sciences, CAS). All data sets were collected under 100 K conditions and integrated using HKL2000[41]. Initial phases were calculated by molecular replacement (MR) in PHASER using a predicted model by AlphaFold2. The final MR solution contains a dimer in the asymmetric unit and resulted in a clear electron density for the nucleotides. Manual building of the models was carried out in COOT[40], following further refinement, adjustment and validation with PHENIX. The diffraction and refinement data were summarized in Supplementary Table 3.

## Relaxation assay

Relaxation activity of pP1192R was assayed by using negatively supercoiled plasmid pUC19 (Solarbio). Briefly, pP1192R (2-1200 nM) was incubated with substrate pUC19 (12 nM) for 30 min at 37°C in a total of 20 µL reaction buffer containing 50 mM Tris−HCl (pH 7.5), 150 mM NaCl, 6 mM $MgCl_2$, 1 mM TCEP, 100 µg/mL BSA and 1 mM ATP. Reaction was stopped by addition of 2 µL 10× DNA loading buffer (Thermo Fisher: BlueJuice™ Gel loading buffer), and samples were directly electrophoresed in a 1.0% (w/v) native agarose gel, with 0.5× TAE buffer (40 mM Tris-Acetate, 1 mM EDTA), for 3 h at 60 V, 4 °C. Supercoiled DNA Ladder Marker was obtained from Takara. Gel was stained with 1× Gelgreen (Gene-star) for 20 min and visualized by UV transillumination (Gel Doc™ EZ Imager, BIO-RAD). At least three independent experiments were carried out.

## Decatenation assay

Decatenation assay was the same as relaxation assay except that pUC19 plasmid was substituted by kinetoplast DNA (kDNA) from *Crithidia fasciculata* (TopoGEN).

## ATPase assay

ATPase activity of the full-length and truncated ATPase domain was determined spectrophotometrically by free inorganic phosphate in solution with Malachite Green Phosphate Detection Kit (Cell Signaling Technology, Inc.). Briefly, 200 nM protein and 500 µM ATP (Solarbio) were incubated in a total of 30 µL reaction buffer (same as Relaxation assay) for 30 min at 37°C. Then, 25 µL reaction mixture was mixed with 100 µL Malachite green reagent as manufacturer's instruction, and absorbance was measured at 630 nm within 5 min. Data were analyzed by OriginPro 2022 software. All reactions were repeated independently at least three times.

## AFM imaging

AFM imaging of pP1192R and DNA interaction was adapted from previous report[7]. A total volume of 20 µL sample (20 nM supercoiled plasmid pUC19 in ddH₂O, 0.5-5 nM pP1192R or pUC19-pP1192R protein complex in 20 mM Hepes (pH 7.5), 1 mM $MgCl_2$, 50 mM NaCl) for AFM was deposited onto freshly cleaved mica (V1, EMS; Pre-treated with 1 mM $MgCl_2$ for pUC19 imaging), and sample was rinsed with 20 µL ultrapure deionized water, then surface was dried using a stream of nitrogen. AFM images were captured using a Dimension FastScan Bio (Bruker) Atomic Force Microscope in ScanAsyst mode with an FastScan-C probe at room temperature at Institute of Process Engineering, CAS.

## EMSA

Binding reactions (20 µL) were carried out in 50 mM Tris−HCl (pH 7.5), 150 mM NaCl, 6 mM $MgCl_2$, 1 mM TCEP, 100 µg/mL BSA (same as relaxation assay except that ATP was omitted). 200 nM 52 bp dsDNA or 100 nM four-way junction DNA (5'-cy5 labeled) was either incubated with 1 µM or 2 µM pP1192R for 30 min at 30 °C. Samples were loaded on a 3-12% native PAGE gel (Thermo Fisher) for electrophoresis at 100 V for 2 h. Gels were imaged by Typhoon FLA 7000, GE Healthcare Bio-Sciences.

## Statistics and reproducibility

The experiments involving relaxation assay, decatenation assay, ATPase assay, and EMSA assay were conducted in independent triplicate for each assay. The data of ATPase assay is presented as values +/− standard error of the mean (SEM) for 3 independent replicates ($n = 3$). Individual data points are also plotted.

## Inhibitor autodock

The SDF structure files of Arctiin was obtained in the PubChem database, and Open Babel[42] (2.4.1) was used to convert SDF into PDB format. The structure of CT1 was obtained from PDB: 8GCC[32]. The crystal structure of the ATPase domain with AMPPNP complex (AMPPNP removed prior to docking) is the receptor for Arctiin. The pP1192R_{CD-DNA} (its binding DNA replaced by 8GCC DNA duplex, whose bending mode is different upon CT1 binding) is the receptor for CT1. The standard docking processes were performed using AutoDock vina[43] (1.1.2). The specific parameters for Arctiin docking are: a grid box size of (15 Å, 15 Å, 15 Å) with a grid center at (19.0, 17.3 and −11.1) for x, y and z respectively. The specific parameters for CT1 docking are: a grid box size of (14 Å, 14 Å, 14 Å) with a grid center at (174.7, 169.2 and 161.8) for x, y and z respectively.

## Figure preparation

All the representing model and cryo-EM density maps were generated using UCSF Chimera[39] and PyMOL[44]. Programs Clustal X[45] and ESPript[46] were used to align multiple sequences.

## Reporting summary

Further information on research design is available in the Nature Portfolio Reporting Summary linked to this article.

# Data availability

The structures of pP1192R and its complexes have been deposited at the Protein Data Bank (PDB). The accession codes are: 8KGR (pP1192R_{CD-DNA}), 8KGO (pP1192R_{Close}), 8KGP (pP1192R_{Coil-open}), 8KGL (pP1192R_{WHD-open}), 8KGQ (pP1192R_{F1}), 8KGM (pP1192R_{F2}), 8KGN (pP1192R_{F3}), 8KGT (ATPase-ADP) and 8KGS (ATPase-AMPPNP). The cryo-EM density maps of pP1192R and its complexes have been deposited at the Electron Microscopy Data Bank. The accession codes are: EMD-37231 (pP1192R_{CD-DNA}), EMD-37228 (pP1192R_{Close}), EMD-37229 (pP1192R_{Coil-open}), EMD-37225 (pP1192R_{WHD-open}), EMD-37230 (pP1192R_{F1}), EMD-37226 (pP1192R_{F2}) and EMD-37227 (pP1192R_{F3}). The previously available coordinates we used for structural analysis include 2RGR (step2 in Fig. 7), 3L4J (step4 in Fig. 7) and 8GCC (CT1). The SDF structure files for Arctiin (https://pubchem.ncbi.nlm.nih.gov/compound/100528) were obtained from the PubChem database.

AlphaFold models used for model building are included in the Source Data file. Additional AFM images have been uploaded to Figshare (https://doi.org/10.6084/m9.figshare.25273231.v1). Source data are provided with this paper.

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

## Acknowledgements

We thank Boling Zhu, Lihong Chen and Xiaojun Huang for cryo-EM data collection, the Center for Biological imaging (CBI) in Institute of Biophysics (IBP) for conducting all EM work. We thank Bei Yang for assistance with our cell biology experiments. We thank Yan Wu for his research assistant service. We thank Nan Wang and Wangjun Fu for their assistant in image processing. We thank Yi Han and Ya Wang for assistance in protein crystal growth and X-ray data collection. We thank Xianjin Ou for technical assistance during the fermentation/protein preparation steps. We also thank Cui Song from Institutional Center for Shared Technologies and Facilities of Institute of Process Engineering, CAS, for necessary characterizations and analysis in AFM. All research described in this article is supported by the Strategic Priority Research Program of the Chinese Academy of Sciences (XDB37030200), National Natural Science Foundation of China (Grant no. 32270179), and Research on the Application of Space Protein Molecular Assembly (YYWT-0901-EXP-10).

## Author contributions

J.C. and Y.X. designed the study. J.C., Y.X., and H.K. performed protein purification and cryo-EM sample preparation. J.C., Y.X., and C.W. performed the cryo-EM data collection and processing. Y.X. and Y.C. performed the X-ray data collection. Y.X., Y.Y. and J.C. conducted biochemical experiments; J.C., Y.X. and Y.C. determined and refined the structure. D.Z. proofread and revised the manuscript. J.C. and Y.X. wrote the manuscript together with X.L. and Y.C. Z.R. supervised the project.

## Competing interests

The authors declare no competing interests.
