## [Peer Review File · Nature Communications]

REVIEWER COMMENTS

Reviewer #1 (Remarks to the Author):

In Structural insights into DNA Topoisomerase II of African Swine Fever Virus, by Cong et al., the authors deepen the knowledge on pP1192R, a unique type II DNA topoisomerase coded by the African Swine Fever virus. Using a combination of cryo-EM and X-ray crystallography the authors determined the protein's structure, and further characterized its activity in vitro by using biochemical assays. These results, especially the structural ones, are important to help understand how this viral enzyme works and to develop protein-drug interaction studies in order to inhibit or control ASFV infection. However, the recent publication of pP1192R's cryo-EM structures (doi: 10.1128/mbio.01228-23), even though without the ATPase/N-gate domain, hinders the novelty of this study. The structures obtained in both cases are highly similar, which strengthens both findings and respective methodologies. Furthermore, Cong et al. went even further and were able to obtain full-length structures, thus having a more complete work. In addition, they also add to what was known about the activity of this protein, in vitro, by showing that it has a binding preference for four-way junctions. This may be relevant to understand the processes for which this protein is essential during ASFV infection.

Given all of this, I have no doubt about the merits of the work and that it deserves to be published. Still, I believe the manuscript requires a revision, improvements and changes should be made and some information added before it is ready for publication.

Issues with the main text:

- line 161, "In addition to the Mg²⁺-binding residues", the authors should indicate which residues they are referring to. Are they the same ones as in line 127? Because Figure 3d shows different ones. Are they the ones mentioned in line 205? Simply mentioning the residues inside brackets should be enough to clarify the reader.

- results from size exclusion chromatography should be explained, since it is not clear at all what the authors intend to show with that Figure (Supplementary Figure 6d) and no explanation is given in the text. Please explain the logic behind the methodology and give a proper interpretation of its results.

- I'm unsure about the authors' interpretation of the AFM data, shown on Supplementary Figure 6. This AFM data looks very different from what is shown in reference #21. I have problems considering that, in Supp Figure 6 a and b, pP1192R is binding preferentially to DNA crossovers, at it can be observed binding different regions of the several DNA molecules present. Perhaps using a lower protein concentration for this assay could yield clearer results and pictures similar to those observed in manuscript reference #21?

- lines 245-246, the authors state "and confirmed its function by in vitro enzymatic assays". The function of this protein was confirmed previously by different authors. I think the authors mean to say that they confirmed that the proteins they purified and for which they obtained the structures were functional.

This is of relevance because it reinsures that the structures were obtained from functional proteins. Therefore, I suggest that the sentence is rephrased to not be misleading.

- In the discussion, the authors dwell on the catalytic cycle of pP1192R, using the structures obtained as support for this but also using structures from the yeast counterpart. I think the explanation of the catalytic cycle could and should be done without the yeast structures. They have enough data to go through different phases of the cycle, and the significant amino acid difference between the two proteins may very well imply structural and functional differences, and hence differences in distances between residues and/or domains, that may hinder the authors' conclusions.

- lines 294-295, "pP1192R has a low similarity on amino acid sequence", but percentage of identity and/or similarity is never indicated throughout the manuscript. These values should either be mentioned here or included in the Figure with the alignment of the 3 topoll sequences (Supplementary Figure 7).

- lines 298-304, it is my opinion that what is stated in this portion of text is very speculative and has no support from the results of the study. And do the authors have some indication as to how or why pA104R would work as a CTD? pA104R is considered to be a histone-like protein and may contribute to compaction of the viral DNA, which is far from what is described for a topoll CTD.

- lines 307-308, I disagree with the authors' claim that "these findings advance our comprehension of the mechanism underlying type IIA topoisomerases". They advance our knowledge of pP1192R as a type II DNA topoisomerase, but not much more than that. What the authors find here is already well know, in even greater depth, for other topolls.

- lines 309-310, authors state that this "work holds the potential to make contributions to the prevention and treatment of African swine fever". This sentence is very generic and without a proper explanation as to why this potential exists or how it could make these contributions, I would refrain from keeping it in the final version of the manuscript. Especially because no results concerning prevention or treatment or shown. Inclusion of docking simulations between topoll inhibitors and the structures obtained, for example, would be a good addition and provide support to their claim.

Concerning the Figures included in the manuscript, I consider that results on the activity of the purified proteins are more relevant than purification details. If, by space concerns, all cannot be shown in the main paper but having the possibility of including them as supplementary data, then I would suggest that Figure 1a could move to Supplementary Figure 1a, Fig 1b would be Fig 1a, Supp Fig 1b would be Fig 1b, and Supp Fig 1a would be Supp Fig 1b. Thus, in Figure 1 the authors would show, in a), the in vitro relaxation and decatenation activity of the purified pP1192R and, in b), the in vitro ATPase activity of the purified full-length pP1192R and truncated ATPase domain.

Additionally, Supplementary Figure 6 includes novel results on the activity of the viral topoll. Therefore, I think that at least panel c, corresponding to the EMSA, should be part of a main figure. In addition, I suggest that the arrows in the AFM panel of this figure point to what the authors consider to be the protein itself, and not its name (i.e., reverse the arrows).

Figure 6 has no indication of being Figure 6. Furthermore, in Step 1 of Figure 6, perhaps the authors want to indicate pP1192Rcoilopen and not pP1192RWHD-open, as it is currently?

Finally, on line 161, authors refer to panels b and c of Supplementary Figure 3, but I don't see how panel c helps support the authors' statement.

Overall, the methodology used is indicated with sufficient detail. I do, however, have some doubts and some suggestions in order to increase the ease with which results may be reproduced:

- in some cases, the supplier or manufacturer for a reagent/consumable/software is indicated, while in other cases it is not; this should be uniformized and indicated for all cases
- line 317, which Bac-to-Bac expression system was used? The authors mention the cell line used, but what was the medium used, growth and expression conditions, MOI used for infection, ...?
- line 333, how were the purified proteins concentrated – devices used, conditions?
- it is not mentioned throughout the paper if the 8xHis tag is removed from the C-terminus of the purified proteins before modelling/after purification, and/or how the tag is not present from the final models (it is not indicated as excluded residues in the supplementary table)
- line 348, how was the purified full-length pP1192R changed into cryo-EM buffer?
- line 423, was the pUC19 used obtained commercially? If so, from which supplier? If not, how was it prepared?
- lines 424-425, why were these conditions chosen to perform relaxation assays with purified pP1192R? They differ from what has been previously published for this protein (manuscript ref #7) and the protein seems to be less active here than observed in manuscript ref #7 (compare relaxation activity when using 5 nM of purified protein, for example)
- line 459, authors mention that binding reactions “were carried out in reaction buffer mentioned above”, but this could be clarified; are the authors referring to the reaction buffer used for the relaxation assays? If so, why not mention it, as done in lines 441-442?
- line 461, authors mention that DNA was incubated with “the indicated amounts of pP1192R”, but these amounts are not indicated in this section nor in the corresponding figure or figure legend (supplementary figure 6).

Some improvements or corrections can be done in terms of references:

- line 35, better references for that statement can be used than ref #3 (there are several good ones from Patrick Forterre et al. or, for a more recent one, doi: 10.1093/ve/veac097)
- lines 49-50, authors mention that “the structure and mechanism of virus-encoded Topo II remained unclear”. This is not precise due to the recent publication of pP1192R's cryo-EM structures, as I've mentioned before. Therefore, this should be rephrased, and that reference added to this paper.
- line 156, reference #3 could be indicated downstream in the sentence, following “...other homologous type IIA topoisomerases”

- line 240, reference #18 does not seem suitable in this context; maybe authors wanted to indicate reference #19?

- line 296, together with references #5 and #16, the paper with doi: 10.1128/mbio.01228-23 should also be mentioned here and added to the list of references included in the manuscript

- line 342, reference should be properly mentioned instead of just "...adapted from Katherine..."

Reviewer #2 (Remarks to the Author):

In this manuscript, the research conducted by Cong et al focuses on determining the structures of pP1192R, the African swine fever virus (ASFV) Topo II enzyme, in different conformational states using cryo-EM, X-ray crystallography, and atomic force microscopy. They also validate the enzymatic activity of pP1192R through biochemical assays. The overall significance of their findings lies in enhancing our understanding of virus-encoded Topo II enzymes and providing potential avenues for intervention strategies against ASFV. However, it is important to note that the novelty of the research may be limited due to a recent publication by Zhao et al in mBIO 2023.

Major

1.The authors have determined the pP1192R structures with distinct conformational states, including pP1192RClose, pP1192RCoil-open and pP1192RWHD-open . It is important to note that there have been previous studies on ASFV P1192R in both the closed and open C-gate forms. Therefore, in order to provide novelty in their findings, the authors should compare their determined structures with those reported in previous studies. By conducting a comparative analysis between the structures obtained in this study and the structures reported in earlier research, the authors can identify any significant differences or unique features that distinguish their work. They can also discuss any new insights or understanding that their structures provide regarding the function or mechanism of ASFV P1192R.

2.Given the lack of commercialized vaccines or drugs for ASFV, it is indeed valuable to discuss the potential for structure-based drug design targeting ASFV P1192R.

Minor

3.Line 24, "confirming"should be confirmed or validated.

4.The author should discuss the reasons why ASFV encodes pP1192R, its own viral topoisomerase, despite the presence of the host cell's Topo II enzyme.

Reviewer #3 (Remarks to the Author):

Major remarks

The introduction is too brief. The state-of-the-art on topoisomerase II should be provided with much more details (structure, mechanisms, ...).

The provided results on the binding of the Topo II are insufficient. The AFM results should be extended considerably. Only 1 AFM image of very poor quality/resolution is provided. The following additional results should be added:

- High-resolution AFM imaging in liquid of the binding of pP1192R to linear DNA, relaxed DNA, and supercoiled DNA (e.g. see reference 33).
- o Quantify DNA-TOPO II particles: binding at one end (linear DNA), duplex, three-way and four-way.
- o Quantify the number of TOPO II on one pUC19 plasmid (e.g. see reference 21)
- o Quantify the crossing angle (four-way DNA)
- o If TOPO II binds to linear DNA, quantify the DNA bending angle (e.g., reference 33). This could confirm the statement on line 154: “..facilitating a global bending and a localized DNA conformational transition ...”
- o Quantify the measured height of the protein on the mica surface and when bound to the DNA (provide also a 3D image of bound protein to the DNA).
- Image the four-way junction DNA complex that was prepared for the EMSA assays and quantify the binding of pP1192R. Compare the results with the ones obtained with pUC19.

These results on the specificity of the binding of pP1192R to DNA should be discussed in the Discussion section.

A comparison of the pP1192R structure to other eukaryotic TOPO 2 (superposition figure) and discussion is missing in the discussion section.

The “Materials and Methods” section is incomplete; many details are missing, e.g.

- Line 421 to 435 and Figure 1b: Relaxation assay: How was pUC19 obtained? Was it purified before use? What was the protein control? What was the negative control? What was the supercoiled DNA marker? A control for relaxed pUC19 should also be included in the gel.

- Line 458, EMSA: Binding reactions were performed at which temperature (at 4°C)? How was the four-way junction DNA prepared? How was the labeling performed? Amounts are not indicated in Suppl. Figure 6c. What were the controls? What represents “Annealed”? What is “O1-Cy5” and “RY-Cy5”?

- See the minor remarks for additional comments.

Minor remarks

Line 135: ... Coiled Coil subdomains => coiled—coil subdomains.

Line 145: ... H3 (755-766), alongside... => ... H3 (755-766) (Fig. 3b), alongside...

Line 146 and 153: “wing-1” and “wing-2” are denoted differently in Fig. 3 and Fig. 3b: “Wing1” and “Wing2”.

Line 147: ... Y800. Specifically, ... => ... Y800 (Supplementary Fig. 3c). Specifically, ...

Line 150: The catalytic Y800 comes into close proximity with the DNA strand (Fig. 3b and Supplementary Fig. 3c). => This is not that clear in Fig. 3b and not shown in Supplementary Fig. 3c. Could this be shown in Suppl. Fig. 3c with an indication of the distance?

Line 157: ...Isoleucine ...=> isoleucine

Line 161: (Supplementary Fig. 3b, c) => (Supplementary Fig. 3b)

Line 164: (2) => ????

Line 164: ... discernible interaction was observed... => Explain in more detail which type of interaction? With which amino acids?

Line 309: “Furthermore, the work holds the potential to make contributions and treatment of African swine fever.” => Explain in which way this could be realized.

Line 342: Katherine?

Figure 2b: Please provide more details in the Materials and Methods on how this figure was obtained.

Figure 6: “Fig. 6” is missing in the caption.

Suppl. Fig.1a: Indicate the molecular weight of the other bands. Do not abbreviate “Lad.”.

Suppl. Fig.1b: What does the error bar represent? What is the “ATP control”? This should be described in Materials and Methods.

Suppl. Fig.1c: Which sample is this?

Suppl. Fig.1d: Which sample is this?

Suppl. Fig. 2: Give more details in the caption. There are 2 subfigures indicated with “f”.

Suppl. Fig. 3a: Include numerical values with the electric potential scale.

Reviewer #4 (Remarks to the Author):

This is a very interesting study of a virally encoded type II topoisomerase, which infects a eukaryote, namely swine. In the same way that the drug targeting of the virally encoded thymidine kinase from Herpes Simplex has led to very successful drugs like Acyclovir, Pencyclovir and Gancyclovir for the treatment of Human herpes infection, one can envisage that treatment of African Swine virus disease in pigs could in the future be treated with a drug (from rational structural drug design) which selectively targets this porcine viral type II topoisomerase. ASFV is a very serious disease which can affect a major food source for the Human population. ASFV has now spread from its original location of Africa to a wide number of other countries and continents like South America.

I very strongly recommend publication of this carefully executed work. Even though there is wide sequence divergence in sequence between this type II DNA topo in ASFV and the Human topoII-alpha and topoII-beta there turns out to be high structural homology. A very long G-gate sequence 52 bp has been used compared with those used in bacterial topoII structural work which is very interesting and a wide range of catalytic states have been teased out of the particle analysis from the cryoEM data using Relion software at high resolution. The comparisons are excellent with other structures which are primarily from bacterial, yeast and human topo II X-ray crystallographic and now increasingly cryoEM studies. In addition an AFM study has been performed as well as the cryoEM, X-ray crystallographic work which greatly adds to the study.

REVIEWER COMMENTS

Reviewer #1 (Remarks to the Author):

In Structural insights into DNA Topoisomerase II of African Swine Fever Virus, by Cong et al., the authors deepen the knowledge on pP1192R, a unique type II DNA topoisomerase coded by the African Swine Fever virus. Using a combination of cryo-EM and X-ray crystallography the authors determined the protein's structure, and further characterized its activity in vitro by using biochemical assays. These results, especially the structural ones, are important to help understand how this viral enzyme works and to develop protein-drug interaction studies in order to inhibit or control ASFV infection. However, the recent publication of pP1192R's cryo-EM structures (doi: 10.1128/mbio.01228-23), even though without the ATPase/N-gate domain, hinders the novelty of this study. The structures obtained in both cases are highly similar, which strengthens both findings and respective methodologies. Furthermore, Cong et al. went even further and were able to obtain full-length structures, thus having a more complete work. In addition, they also add to what was known about the activity of this protein, in vitro, by showing that it has a binding preference for four-way junctions. This may be relevant to understand the processes for which this protein is essential during ASFV infection.

Given all of this, I have no doubt about the merits of the work and that it deserves to be published. Still, I believe the manuscript requires a revision, improvements and changes should be made and some information added before it is ready for publication.

Our response: We thank the reviewer for the summary and suggestions for our work. We have revised the presentation with reference to the suggestions.

Issues with the main text:

- line 161, "In addition to the Mg²⁺-binding residues", the authors should indicate which residues they are referring to. Are they the same ones as in line 127? Because Figure 3d shows different ones. Are they the ones mentioned in line 205? Simply mentioning the residues inside brackets should be enough to clarify the reader.

Our response: We thank the reviewer for suggesting this improvement. We have included the residues (E438, D539, and D541) after the sentence as suggested. The residues mentioned in line 127 should be the same as in line 205 and Figure 3d, and we have rectified this error.

- results from size exclusion chromatography should be explained, since it is not clear at all what the authors intend to show with that Figure (Supplementary Figure 6d) and no explanation is given in the text. Please explain the logic behind the methodology and give a proper interpretation of its results.

Our response: We appreciate the reviewer's suggestion for this improvement. We intended to use changes in the 260/280 ratio (from 0.36 to 1.55) as evidence to support the binding between the protein and DNA. We have included the explanation in the manuscript and legend.

- I'm unsure about the authors' interpretation of the AFM data, shown on Supplementary Figure 6. This AFM data looks very different from what is shown in reference #21. I have problems considering that, in Supp Figure 6 a and b, pP1192R is binding preferentially to DNA crossovers,

at it can be observed binding different regions of the several DNA molecules present. Perhaps using a lower protein concentration for this assay could yield clearer results and pictures similar to those observed in manuscript reference #21?

Our response: We appreciate the suggestions from the reviewer and have now re-done the AFM experiment according to the reviewer's recommendations. The new results are added to Figure 6 and the manuscript has been updated with the relevant changes.

- lines 245-246, the authors state "and confirmed its function by in vitro enzymatic assays". The function of this protein was confirmed previously by different authors. I think the authors mean to say that they confirmed that the proteins they purified and for which they obtained the structures were functional. This is of relevance because it reinsures that the structures were obtained from functional proteins. Therefore, I suggest that the sentence is rephrased to not be misleading.

Our response: We appreciate the suggestions from the reviewer and have rephrased the sentence "and confirmed that the proteins we purified are functional through in vitro enzymatic assays".

- In the discussion, the authors dwell on the catalytic cycle of pP1192R, using the structures obtained as support for this but also using structures from the yeast counterpart. I think the explanation of the catalytic cycle could and should be done without the yeast structures. They have enough data to go through different phases of the cycle, and the significant amino acid difference between the two proteins may very well imply structural and functional differences, and hence differences in distances between residues and/or domains, that may hinder the authors' conclusions.

Our response: We thank the reviewer for suggesting this improvement. Our main focus is on whether the distance can support nucleophilic attack. The inclusion of yeast structures is intended to make this process more comprehensive and detailed. We only compared the distance between the catalytic tyrosine and the nucleic acid, which is highly conserved among various Topo IIs. The measurement of these distances are accurate, which could be a criteria to classify the state of Topo II in DNA cleavage process. In order to make a better distinction we have changed the color of the yeast catalytic tyrosine to yellow. Furthermore, we have revised our manuscript and included a comparison between pP1192R and yeast Topo II in supplementary Figure 7 c,d .

- lines 294-295, "pP1192R has a low similarity on amino acid sequence", but percentage of identity and/or similarity is never indicated throughout the manuscript. These values should either be mentioned here or included in the Figure with the alignment of the 3 topo II sequences (Supplementary Figure 7).

Our response: We appreciate the suggestions from the reviewer and have incorporated the percentage of identity in the figure legend of Supplementary Figure 9.

- lines 298-304, it is my opinion that what is stated in this portion of text is very speculative and has no support from the results of the study. And do the authors have some indication as to how or why pA104R would work as a CTD? pA104R is considered to be a histone-like protein and may contribute to compaction of the viral DNA, which is far from what is described for a topoII CTD.

Our response: We appreciate the suggestions from the reviewer and have included an additional reference (doi: 10.1073/pnas.93.25.14416.) which indicates that deletion of the C-terminal gyrase

results in the inability to supercoil DNA, instead causing relaxation of DNA. The original #24 showed that pA104R cooperates with ASFV topoisomerase II (pP1192R) to modulate DNA supercoiling. As a result, we hypothesized that pA104R could function similarly to the CTD of gyrase, which is indeed our speculation. We hope that by presenting this hypothesis here, it will contribute to the future exploration of other researchers.

- lines 307-308, I disagree with the authors' claim that "these findings advance our comprehension of the mechanism underlying type IIA topoisomerases". They advance our knowledge of pP1192R as a type II DNA topoisomerase, but not much more than that. What the authors find here is already well known, in even greater depth, for other topoIIs.

Our response: We appreciate the suggestions from the reviewer we rephased "these findings advance our comprehension of the mechanism underlying viral type IIA topoisomerases"

- lines 309-310, authors state that this "work holds the potential to make contributions to the prevention and treatment of African swine fever". This sentence is very generic and without a proper explanation as to why this potential exists or how it could make these contributions, I would refrain from keeping it in the final version of the manuscript. Especially because no results concerning prevention or treatment are shown. Inclusion of docking simulations between topoII inhibitors and the structures obtained, for example, would be a good addition and provide support to their claim.

Our response: We thank the reviewer for suggesting this improvement. We have docked two topo II inhibitors in the ATPase domain and central domain of pP1192R. These results have been included in the supplementary Figure 8, and a discussion section has been added to address these findings.

Concerning the Figures included in the manuscript, I consider that results on the activity of the purified proteins are more relevant than purification details. If, by space concerns, all cannot be shown in the main paper but having the possibility of including them as supplementary data, then I would suggest that Figure 1a could move to Supplementary Figure 1a, Fig 1b would be Fig 1a, Supp Fig 1b would be Fig 1b, and Supp Fig 1a would be Supp Fig 1b. Thus, in Figure 1 the authors would show, in a), the in vitro relaxation and decatenation activity of the purified pP1192R and, in b), the in vitro ATPase activity of the purified full-length pP1192R and truncated ATPase domain.

Our response: We appreciate the suggestions from the reviewer and have made modifications accordingly.

Additionally, Supplementary Figure 6 includes novel results on the activity of the viral topoII. Therefore, I think that at least panel c, corresponding to the EMSA, should be part of a main figure. In addition, I suggest that the arrows in the AFM panel of this figure point to what the authors consider to be the protein itself, and not its name (i.e., reverse the arrows).

Our response: We thank the reviewer for suggesting this improvement. We have changed the EMSA and AFM images into the main figure and corrected the arrows.

Figure 6 has no indication of being Figure 6. Furthermore, in Step 1 of Figure 6, perhaps the authors want to indicate pP1192Rcoilopen and not pP1192RWHD-open, as it is currently?

Our response: We thank the reviewer for pointing out this error, and we have made the correction.

Finally, on line 161, authors refer to panels b and c of Supplementary Figure 3, but I don't see how panel c helps support the authors' statement.

Our response: We thank suggestion from the reviewer and we have removed the reference to panel c.

Overall, the methodology used is indicated with sufficient detail. I do, however, have some doubts and some suggestions in order to increase the ease with which results may be reproduced:

- in some cases, the supplier or manufacturer for a reagent/consumable/software is indicated, while in other cases it is not; this should be uniformized and indicated for all cases

Our response: We appreciate the reviewer's suggestion and have included the respective supplier or manufacturer for a reagent/consumable/software that was previously missing.

- line 317, which Bac-to-Bac expression system was used? The authors mention the cell line used, but what was the medium used, growth and expression conditions, MOI used for infection, ...?

Our response: We thank suggestions from the reviewer and have included the catalog numbers of the Bac-to-Bac expression system (Invitrogen catalog no: 10359-016) in method section (all the necessary details provided in the protocol on the Invitrogen website including the medium used, growth and expression conditions). Furthermore, we have MOI employed during the experiment.

- line 333, how were the purified proteins concentrated – devices used, conditions?

Our response: We thank suggestions from the reviewer. The purified proteins were concentrated at 4 °C through centrifugation at 2,600 g using a centrifugal filter (Merck Millipore) with a molecular weight cutoff of 30 kD to a concentration of 5-15 mg/ml for crystallization. We have incorporated the details of the concentration process in the method section.

- it is not mentioned throughout the paper if the 8xHis tag is removed from the C-terminus of the purified proteins before modelling/after purification, and/or how the tag is not present from the final models (it is not indicated as excluded residues in the supplementary table)

Our response: We did not remove the 8xHis tag in our experiments. The reason for its absence in the final models is due to its location at the terminal end of the protein and its flexibility, therefore its density is missing in the solved cryo-EM map.

- line 348, how was the purified full-length pP1192R changed into cryo-EM buffer?

Our response: We thank suggestions from the reviewer and we have incorporated the dilute buffer (50 mM Hepes (pH 7.2), 4 mM Mg²⁺ and 2 mM TCEP) in the method section. We mixed the purified protein with the dilute buffer at a 1:1 ratio, thereby converting the protein into the cryo-EM buffer.

- line 423, was the pUC19 used obtained commercially? If so, from which supplier? If not, how was it prepared?

Our response: We thank suggestions from the reviewer. The pUC19 was obtained commercially from Solarbio and we have now included the manufacturer's information of pUC19.

- lines 424-425, why were these conditions chosen to perform relaxation assays with purified pP1192R? They differ from what has been previously published for this protein (manuscript ref #7) and the protein seems to be less active here than observed in manuscript ref #7 (compare relaxation activity when using 5 nM of purified protein, for example)

Our response: The buffer of ref #7 is 50 mM Tris-HCl pH 7.5, 100 mM NaCl, 10 mM MgCl₂, 2 mM ATP and 1 mM DTT. The buffer we chose is 50 mM Tris-HCl (pH 7.5), 150 mM NaCl, 6 mM MgCl₂, 1 mM TCEP, 100 µg/mL BSA and 1 mM ATP. The reason we chose the buffer we used was because the concentration of 150mM NaCl is closer to the physiological condition. We chose 6mM MgCl₂ is based on ref #7 Fig. 5a, where the activity of pP1192R is optimal under this condition. The addition of BSA is aimed at protecting purified pP1192R, as in ref #5. Our pP1192R and ref#7 are both able to completely relax all supercoiled plasmids at a concentration of 5nM pP1192R. The lower activity of our protein could be due to the difference in DNA substrate used. They used pRYG plasmid, which is a negative-supercoiled plasmid designed for Topo II (ref: Nucleic Acids Res. 1990 Jan 11;18(1):1-11. doi: 10.1093/nar/18.1.1.). Additionally, the initial ATP concentration is twice as high as ours, although theoretically, concentrations of 1-2mM are saturating for this reaction.

- line 459, authors mention that binding reactions “were carried out in reaction buffer mentioned above”, but this could be clarified; are the authors referring to the reaction buffer used for the relaxation assays? If so, why not mention it, as done in lines 441-442?

Our response: We thank suggestions from the reviewer and we have now included the EMSA buffer in method section.

- line 461, authors mention that DNA was incubated with “the indicated amounts of pP1192R”, but these amounts are not indicated in this section nor in the corresponding figure or figure legend (supplementary figure 6).

Our response: We thank reviewer for suggesting this improvement. we have added the information in figure legend.

Some improvements or corrections can be done in terms of references:

- line 35, better references for that statement can be used than ref #3 (there are several good ones from Patrick Forterre et al. or, for a more recent one, doi: 10.1093/ve/veac097)

Our response: We appreciate the reviewer's suggestion and have replaced the reference.

- lines 49-50, authors mention that “the structure and mechanism of virus-encoded Topo II remained unclear”. This is not precise due to the recent publication of pP1192R's cryo-EM structures, as I've mentioned before. Therefore, this should be rephrased, and that reference added to this paper.

Our response: We appreciate the suggestion from the reviewer. At the time of the manuscript

submission, the paper by Zhao et al was unpublished. We have now incorporated the reference in the sentence and revised its phrasing to ensure clarity and accuracy.

- line 156, reference #3 could be indicated downstream in the sentence, following “...other homologous type IIA topoisomerases”

Our response: As the reviewer’s suggestion, we have repositioned the original reference #3 to the correct position.

- line 240, reference #18 does not seem suitable in this context; maybe authors wanted to indicate reference #19?

Our response: We thank the reviewer for pointing out this error, and we have made the correction.

- line 296, together with references #5 and #16, the paper with doi: 10.1128/mbio.01228-23 should also be mentioned here and added to the list of references included in the manuscript

Our response: We appreciate the suggestion from the reviewer. At the time of the manuscript submission, the paper by Zhao et al was unpublished. We have now incorporated the reference.

- line 342, reference should be properly mentioned instead of just “...adapted from Katherine...”

Our response: We appreciate the suggestion from the reviewer and we have rephrased the sentence.

Reviewer #2 (Remarks to the Author):

In this manuscript, the research conducted by Cong et al focuses on determining the structures of pP1192R, the African swine fever virus (ASFV) Topo II enzyme, in different conformational states using cryo-EM, X-ray crystallography, and atomic force microscopy. They also validate the enzymatic activity of pP1192R through biochemical assays. The overall significance of their findings lies in enhancing our understanding of virus-encoded Topo II enzymes and providing potential avenues for intervention strategies against ASFV. However, it is important to note that the novelty of the research may be limited due to a recent publication by Zhao et al in mBIO 2023.

Our response: We thank the reviewer for the summary and suggestions for our work. We have revised the presentation with reference to the suggestions.

Major

1.The authors have determined the pP1192R structures with distinct conformational states, including pP1192RClose, pP1192RCoil-open and pP1192RWHD-open . It is important to note that there have been previous studies on ASFV P1192R in both the closed and open C-gate forms. Therefore, in order to provide novelty in their findings, the authors should compare their determined structures with those reported in previous studies. By conducting a comparative analysis between the structures obtained in this study and the structures reported in earlier research, the authors can identify any significant differences or unique features that distinguish their work. They can also discuss any new insights or understanding that their structures provide regarding the function or mechanism of ASFV P1192R.

Our response: We appreciate the suggestions made by the reviewer. At the time of the manuscript

submission, the paper by Zhao et al was not published. We have compared our structure with the published structures (PDB: 8J9Y, 8J9Z) and included these in the discussion section. The structures are nearly identical, with RMSD values of 0.77 and 0.73 Å, respectively. The similarity between the structures obtained in both cases cross-validate our findings. Importantly, our work extends beyond the conformational change in the apo central domain, and the discussion about the two states only occupies a small portion of our paper. Therefore, in comparison to Zhao's work, our research offers a more comprehensive analysis. In terms of structure, we not only determined the structure of the apo central domain, but also successfully solved the full-length structures in different conformations. Furthermore, we incorporated the substrate components (DNA, ATP, AMPPNP, Mg²⁺) and we investigated to provide insights into its mechanism by multiple conformations of pP1192R. In regards to functionality, we employed enzymatic activity, AFM (supercoiled DNA) and EMSA (four-way junction DNA) experiments to explore the natural binding modes.

2. Given the lack of commercialized vaccines or drugs for ASFV, it is indeed valuable to discuss the potential for structure-based drug design targeting ASFV P1192R.

Our response: We thank the reviewer for suggesting this improvement. We have docked two topoisomerase II inhibitors in the ATPase domain and central domain of pP1192R. These results have been included in the supplementary Figure 8, and a discussion section has been added to address these findings.

Minor

3. Line 24, "confirming" should be confirmed or validated.

Our response: We thank the reviewer for pointing out this error, and we have made the correction.

4. The author should discuss the reasons why ASFV encodes pP1192R, its own viral topoisomerase, despite the presence of the host cell's Topo II enzyme.

Our response: We thank the reviewer for suggesting this improvement. ASFV encoding its own Topo II could be attributed to the fact that its large genomic DNA is prone to tangling. Besides, DNA replication of ASFV occurs in cytoplasmic factories, while the host's topoisomerase II mainly exists in the nucleus. By encoding its own Topo II, the virus may efficiently replicate within the host cell. We have included our hypothesis regarding the reasons in the discussion section of the second-to-last paragraph.

Reviewer #3 (Remarks to the Author):

Major remarks

The introduction is too brief. The state-of-the-art on topoisomerase II should be provided with much more details (structure, mechanisms, ...).

Our response: We thank the reviewer for suggesting this improvement. We have rephrased the introduction, providing a description of the structural configuration of topoisomerase II and highlighted its key catalytic sites. We have summarized mechanisms of topoisomerase II including the

cleavage of the G-segment, passage of the T-segment, and resealing of the G-segment to accomplish the relaxation/decatenation process. Furthermore, we outline the function of topo II in eukaryotes, bacteria, and viruses.

The provided results on the binding of the Topo II are insufficient. The AFM results should be extended considerably. Only 1 AFM image of very poor quality/resolution is provided. The following additional results should be added:

- High-resolution AFM imaging in liquid of the binding of pP1192R to linear DNA, relaxed DNA, and supercoiled DNA (e.g. see reference 33).

Our response: We thank the reviewer for suggesting this improvement. We have provide more AFM results in Figure 6 and supplementary Figure 6. However, we would like to emphasize that the focus of this paper is the structure of pP1192R, and the purpose of the AFM experiments was simply to prove that pP1192R has a preference for binding at DNA crossovers. The AFM images of pP1192R binding to supercoiled DNA is sufficient to support this conclusion. The protein in reference 33, spo11 binds to DNA ends with high affinity, which is quite different from pP1192R. As suggested, we also performed experiments with linear and relaxed DNA, we found that the majority of proteins in these samples did not bind. As these images are not necessary to support our main point, we have chosen not to include them in the Figure.

o Quantify DNA-TOPO II particles: binding at one end (linear DNA), duplex, three-way and four-way.

We thank the reviewer for suggesting this improvement. However, we would like to clarify that the protein mentioned in reference 33, Spo11, exhibits high affinity binding to DNA ends, which is quite different from pP1192R. Thus we cannot quantify DNA-Topo II particles binding at one end and three-way. We have quantified DNA-Topo II particles binding to duplex and crossover in Figure 6.

o Quantify the number of TOPO II on one pUC19 plasmid (e.g. see reference 21)

Our response: We thank the reviewer for suggesting this improvement. We have quantified the number of Topo II on one pUC19 plasmid. Please note that the reference 21 used pPBR322, which is ~4300 bp in size, larger than pUC19 (~2700 bp). Because of its larger size, a single pPBR322 may bind more proteins than pUC19.

o Quantify the crossing angle (four-way DNA)

Our response: We thank the reviewer for suggesting this improvement. We have quantified the crossing angles in Figure 6.

o If TOPO II binds to linear DNA, quantify the DNA bending angle (e.g., reference 33). This could confirm the statement on line 154: “..facilitating a global bending and a localized DNA conformational transition ...”

Our response: We thank the reviewer for pointing this out. The majority of pP1192R in AFM samples did not bind to linear DNA, therefore there is no statistical significance. We have added the bending angle of pP1192R by analyzing the DNA duplex in our structure. We believe that the structure solved using cryo-EM provides more accurate and precise information compared to

AFM.

o Quantify the measured height of the protein on the mica surface and when bound to the DNA (provide also a 3D image of bound protein to the DNA).

Our response: We thank the reviewer for suggesting this improvement. The average height of the protein on the mica surface is ~3 nm, and we have provided the 3D image in Figure 6.

- Image the four-way junction DNA complex that was prepared for the EMSA assays and quantify the binding of pP1192R. Compare the results with the ones obtained with pUC19.

Our response: We thank suggestions from the reviewer. However, it should be noted that the size of the four-way junction observed in AFM is similar to that of pP1192R, both being approximately 30 nm. Therefore, it is not feasible to use AFM to image the complex formed by the four-way junction DNA - pP1192R.

These results on the specificity of the binding of pP1192R to DNA should be discussed in the Discussion section.

Our response: We thank the reviewer for suggesting this improvement. We have added this in discussion section.

A comparison of the pP1192R structure to other eukaryotic TOPO 2 (superposition figure) and discussion is missing in the discussion section.

Our response: We thank the reviewer for suggesting this improvement. We performed a superposition of the pP1192R structure with that of *S. cerevisiae* (2RGR and 3L4J), respectively. The results have been incorporated into supplementary Figure 7 c, d.

The “Materials and Methods” section is incomplete; many details are missing, e.g.

- Line 421 to 435 and Figure 1b: Relaxation assay: How was pUC19 obtained? Was it purified

before use? What was the protein control? What was the negative control? What was the supercoiled DNA marker? A control for relaxed pUC19 should also be included in the gel.

Our response: We thank suggestions from the reviewer and we have now included the manufacturer's information of pUC19. We have rephrased "protein control" to "protein only" and "negative control" to "pUC19 only". The supercoiled DNA marker is a marker including 8 different supercoiled DNA produced by Takara (#3585A), which we have added in the method. Relaxed pUC19 plasmid is commercially unavailable. Therefore it is not included in the relaxation assay, as in reference 2,7,8. Different plasmid topoisomers indicate they are relaxed.

- Line 458, EMSA: Binding reactions were performed at which temperature (at 4°C)? How was the four-way junction DNA prepared? How was the labeling performed? Amounts are not indicated in Suppl. Figure 6c. What were the controls? What represents "Annealed"? What is "O1-Cy5" and "RY-Cy5"?

Our response: We thank the reviewer for suggesting this improvement. The labeling was performed by manufacturer BGI. The preparation procedure followed the previous study (West KL, Austin CA. Human DNA topoisomerase II beta binds and cleaves four-way junction DNA in vitro. Nucleic acids research 27, 984-992 (1999)) . And we have rephrased the results of EMSA for clarification.

- See the minor remarks for additional comments.

Minor remarks

Line 135: ... Coiled Coil subdomains => coiled—coil subdomains.

Our response: We thank suggestions from the reviewer and we have replaced Coiled Coil subdomains to Coiled-Coil subdomains.

Line 145: ... H3 (755-766), alongside... => ... H3 (755-766) (Fig. 3b), alongside...

Our response: We thank suggestions from the reviewer and we have moved the (Fig.3b).

Line 146 and 153: "wing-1" and "wing-2" are denoted differently in Fig. 3 and Fig. 3b: "Wing1" and "Wing2".

Our response: We thank suggestions from the reviewer and we have replace "Wing1" and "Wing2" to "wing-1" and "wing-2".

Line 147: ... Y800. Specifically, ... => ... Y800 (Supplementary Fig. 3c). Specifically, ...

Our response: We thank suggestions from the reviewer and we have added the (Supplementary Fig. 3c).

Line 150: The catalytic Y800 comes into close proximity with the DNA strand (Fig. 3b and Supplementary Fig. 3c). => This is not that clear in Fig. 3b and not shown in Supplementary Fig. 3c. Could this be shown in Suppl. Fig. 3c with an indication of the distance?

Our response: We thank the reviewer for suggesting this improvement. we have shown the distance between Y800 and DNA in supplementary Fig.3c.

Line 157: ...Isoleucine ...=> isoleucine

Our response: We thank the reviewer for pointing out this error, and we have made the correction.

Line 161: (Supplementary Fig. 3b, c) => (Supplementary Fig. 3b)

Our response: We thank the reviewer for pointing out this error, and we have made the correction.

Line 164: (2) => ????

Our response: The “(2)” in line 164 is corresponding to line 142 “This intricate interaction can be categorized into two distinct regions: (1) Within the central domain..... (2) Within the ATPase domain.....”

Line 164: ... discernible interaction was observed... => Explain in more detail which type of interaction? With which amino acids?

Our response: We appreciate the suggestion made by the reviewer. However, the resolution of pP1192R_{FI} is 5.6 Å, which restricts our ability to observe the intricate details of the interaction. We can only identify a discernible interaction area by analyzing the density map.

Line 309: “Furthermore, the work holds the potential to make contributions and treatment of African swine fever.” => Explain in which way this could be realized.

Our response: We thank the reviewer for suggesting this improvement. We have docking two topo II inhibitors in the ATPase domain and central domain of pP1192R. The results of these have been included in the supplementary Figure 8, and a discussion section has been added to address these findings.

Line 342: Katherine?

Our response: We appreciate the suggestion from the reviewer and we have rephrased the sentence.

Figure 2b: Please provide more details in the Materials and Methods on how this figure was obtained.

Our response: We appreciate the suggestion from the reviewer and we have provided the details in the methods section.

Figure 6: “Fig. 6” is missing in the caption.

Our response: We thank the reviewer for pointing out this error, and we have added “Fig.7” in the caption accordingly.

Suppl. Fig.1a: Indicate the molecular weight of the other bands. Do not abbreviate “Lad.”.

Our response: We thank the reviewer for pointing out this and we have changed “Lad.” to “Marker” and indicated all the molecular weight of the bands.

Suppl. Fig.1b: What does the error bar represent? What is the “ATP control”? This should be described in Materials and Methods.

Our response: We thank the reviewer for suggesting this improvement. This error bar represents the standard deviation of three repeated experiments. We have rephrased “ATP control” to “ATP only” .

Suppl. Fig.1c: Which sample is this

Our response: We appreciate the suggestion from the reviewer. This sample is apo full-length pP1192R and we have added the sample information in figure legend.

Suppl. Fig.1d: Which sample is this?

Our response: We appreciate the suggestion from the reviewer. This sample is apo full-length pP1192R and we have added the sample information in figure legend.

Suppl. Fig. 2: Give more details in the caption. There are 2 subfigures indicated with “F”.

Our response: We have made the improvements by adding the details in the figure caption. Additionally, we have also corrected the mentioned “F”.

Suppl. Fig. 3a: Include numerical values with the electric potential scale.

Our response: We appreciate the suggestion from the reviewer and we have added the numerical values with the electric potential scale.

Reviewer #4 (Remarks to the Author):

This is a very interesting study of a virally encoded type II topoisomerase, which infects a eukaryote, namely swine. In the same way that the drug targeting of the virally encoded thymidine kinase from Herpes Simplex has led to very successful drugs like Acyclovir, Pencyclovir and Gancyclovir for the treatment of Human herpes infection, one can envisage that treatment of African Swine virus disease in pigs could in the future be treated with a drug (from rational structural drug design) which selectively targets this porcine viral type II topoisomerase. ASFV is a very serious disease which can affect a major food source for the Human population. ASFV has now spread from its original location of Africa to a wide number of other countries and continents like South America.

I very strongly recommend publication of this carefully executed work. Even though there is wide sequence divergence in sequence between this type II DNA topo in ASFV and the Human topoII-alpha and topoII-beta there turns out to be high structural homology. A very long G-gate sequence 52 bp has been used compared with those used in bacterial topoII structural work which is very interesting and a wide range of catalytic states have been teased out of the particle analysis from the cryoEM data using Relion software at high resolution. The comparisons are excellent with other structures which are primarily from bacterial, yeast and human topo II X-ray crystallographic and now increasingly cryoEM studies. In addition an AFM study has been performed as well as the cryoEM, X-ray crystallographic work which greatly adds to the study.

Our response: We express our gratitude to the reviewer for his/her concise summary of our main findings and positive feedback. We sincerely appreciate the insightful suggestion, which prompted us to conduct docking simulations to explore potential inhibitors for pP1192R. We have included

the results of these simulations in supplementary Figure 8 and discussed them in detail in the discussion section.

REVIEWER COMMENTS

Reviewer #1 (Remarks to the Author):

I thank the authors for the work they put into reviewing and improving the manuscript and responding to all the issues I and most of the other reviewers raised. It really improved their work, which is now acceptable for publication.

I just have a few minor comments about the reviewed version of the manuscript:

- I welcome the addition of docking simulations of two topo II inhibitors, namely arctiin and CT1. However, CT1 has not been tested on pP1192R, while other compounds have been shown to have an effect on it (ref. 7). As proof of concept, a working and a non-working compound, preferably acting on the same domain, could have been docked to show the difference(s) between them and why one has an effect and the other one doesn't.
- in the Introduction (lines 40-42), the phrase "In bacteria, they encode a unique Topo II called gyrase, which is a heterotetramer and has ability to introduce negative supercoils into closed-circular DNA." can be misinterpreted and a non-expert may think that bacteria only encode DNA gyrase, which is incorrect. This sentence should be clarified.
- lines 314-315, the sentence "The results of AFM [...] bind the crossover site." needs improvement.

Reviewer #2 (Remarks to the Author):

The Authors have addressed all of my concerns with the original manuscript.

Reviewer #4 (Remarks to the Author):

The manuscript is much improved following the reviewers comments. The drug docking studies are a very good enhancement and interesting enhancement to the paper. My recommendation from the first review stands.

REVIEWER COMMENTS to the revised manuscript

Reviewer #3:

Major remarks

The introduction is too brief. The state-of-the-art on topoisomerase II should be provided with much more details (structure, mechanisms, ...).

Our response: We thank the reviewer for suggesting this improvement. We have rephrased the introduction, providing a description of the structural configuration of topoisomerase II and highlighted its key catalytic sites. We have summarized mechanisms of topo II including the cleavage of the G-segment, passage of the T-segment, and resealing of the G-segment to accomplish the relaxation/decatenation process. Furthermore, we outline the function of topo II in eukaryotes, bacteria, and viruses.

Remarks reviewer:

The provided additional information is still limited in situating the topoisomerase II of viruses in the subfamilies of Type II topoisomerases. Besides the presence in bacteria, eukaryotic cells, and viruses (as mentioned), Topo IIs can also be present in archaea, plants, and certain algae. In what aspects do Topo IIA differ from Type IIB topoisomerases?

The provided results on the binding of the Topo II are insufficient. The AFM results should be extended considerably. Only 1 AFM image of very poor quality/resolution is provided. The following additional results should be added:

- High-resolution AFM imaging in liquid of the binding of pP1192R to linear DNA, relaxed DNA, and supercoiled DNA (e.g. see reference 33).

Our response: We thank the reviewer for suggesting this improvement. We have provide more AFM results in Figure 6 and supplementary Figure 6. However, we would like to emphasize that the focus of this paper is the structure of pP1192R, and the purpose of the AFM experiments was simply to prove that pP1192R has a preference for binding at DNA crossovers. The AFM images of pP1192R binding to supercoiled DNA is sufficient to support this conclusion. The protein in reference 33, spo11 binds to DNA ends with high affinity, which is quite different from pP1192R. As suggested, we also performed experiments with linear and relaxed DNA, we found that the majority of proteins in these samples did not bind. As these images are not necessary to support our main point, we have chosen not to include them in the Figure.

Liner DNA with pP1192R

Relaxed DNA with pP1192R

o Quantify DNA-TOPO II particles: binding at one end (linear DNA), duplex, three-way and four-way.

We thank the reviewer for suggesting this improvement. However, we would like to clarify that the protein mentioned in reference 33, Spo11, exhibits high affinity binding to DNA ends, which is quite different from pP1192R. Thus we cannot quantify DNA-Topo II particles binding at one end and three-way. We have quantified DNA-Topo II particles binding to duplex and crossover in Figure 6.

o Quantify the number of TOPO II on one pUC19 plasmid (e.g. see reference 21)

Our response: We thank the reviewer for suggesting this improvement. We have quantified the number of Topo II on one pUC19 plasmid. Please note that the reference 21 used pPBR322, which is ~4300 bp in size, larger than pUC19 (~2700 bp). Because of its larger size, a single pPBR322 may bind more proteins than pUC19.

o Quantify the crossing angle (four-way DNA)

Our response: We thank the reviewer for suggesting this improvement. We have quantified the crossing angles in Figure 6.

o If TOPO II binds to linear DNA, quantify the DNA bending angle (e.g., reference 33). This could confirm the statement on line 154: “..facilitating a global bending and a localized DNA conformational transition ...”

Our response: We thank the reviewer for pointing this out. The majority of pP1192R in AFM samples did not bind to linear DNA, therefore there is no statistical significance. We have added the bending angle of pP1192R by analyzing the DNA duplex in our structure. We believe that the structure solved using cryo-EM provides more accurate and precise information compared to AFM.

o Quantify the measured height of the protein on the mica surface and when bound to the DNA (provide also a 3D image of bound protein to the DNA).

Our response: We thank the reviewer for suggesting this improvement. The average height of the protein on the mica surface is ~3 nm, and we have provided the 3D image in Figure 6.

- Image the four-way junction DNA complex that was prepared for the EMSA assays and quantify the binding of pP1192R. Compare the results with the ones obtained with pUC19.

Our response: We thank suggestions from the reviewer. However, it should be noted that the size of the four-way junction observed in AFM is similar to that of pP1192R, both being approximately 30 nm. Therefore, it is not feasible to use AFM to image the complex formed by the four-way junction DNA - pP1192R.

Remarks reviewer:

Besides visualization, AFM can be used to obtain mechanistic insights and this should be improved.

In the provided pictures (Fig. 6), it can be observed that pP1192R can – besides binding to a cross-over - also bind to the linear DNA parts of the supercoiled plasmid.

Fig. 6b is not that clear. Fig. 6c is fine to better see that there is binding (the tilting angle should be increased, and the Z range reduced). Can this be done for all images in Fig. 6b? Also, it should be clearly shown how many pP119R bind to the plasmid (see e.g. [10.1016/j.febslet.2011.08.051](https://doi.org/10.1016/j.febslet.2011.08.051)). A more detailed discussion should be added to the text (compare the results with other studies).

Fig. 6d, left panel: Show the additional AFM images as Supplementary Material that support these results. Could a bending of the DNA be observed for the duplex case? If so, make a graph “Frequency versus angle (0-180°)”. If not, show a few cases (3D image zoomed in) showing no bending. This figure should be discussed in the manuscript.

Fig. 6d, right panel: This figure should be discussed in the manuscript.

Also, the results on the relaxed plasmids show that pP1192R can bind to linear DNA. This can be added as Supplementary Material.

Supplementary Fig. 6a: Give also cross sections at a few locations supporting the measured average height.

Supplementary Fig. 6b: It seems that aggregation is present. Give a few cross-sections for the non-aggregated protein. Also give some cross-section when the

protein is bound to the DNA (cross-over, linear) supporting the measured height of DNA and protein bound.

Supplementary Fig. 6c: Remove the “0” bar chart and give a graph “probability versus the number of Topo II on the plasmid” (see e.g. [10.1016/j.febslet.2011.08.051](https://doi.org/10.1016/j.febslet.2011.08.051)). Discuss these graphs in the manuscript.

Image the four-way junction DNA complex: The shown image demonstrates that the sample has to be diluted. Nevertheless, larger four-way junctions could have been made (see e.g. [10.1016/j.febslet.2011.08.051](https://doi.org/10.1016/j.febslet.2011.08.051)).

REVIEWER COMMENTS

Reviewer #1 (Remarks to the Author):

I thank the authors for the work they put into reviewing and improving the manuscript and responding to all the issues I and most of the other reviewers raised. It really improved their work, which is now acceptable for publication.

Our response: We appreciate the reviewer's validation and have made revisions to the manuscript based on the following comments.

I just have a few minor comments about the reviewed version of the manuscript:

- I welcome the addition of docking simulations of two topo II inhibitors, namely arctiin and CT1. However, CT1 has not been tested on pP1192R, while other compounds have been shown to have an effect on it (ref. 7). As proof of concept, a working and a non-working compound, preferably acting on the same domain, could have been docked to show the difference(s) between them and why one has an effect and the other one doesn't.

Our response: We appreciate the reviewer's comment, and we would like to provide some clarification regarding our choice of these two drugs. The reason for selecting one as an ATPase domain inhibitor and the other as a Central domain inhibitor was to demonstrate different approaches to inhibition. We acknowledge that there is uncertainty about the inhibitory effect of CT1, and mentioning it was intended only to illustrate that it could serve as a reference for deriving new compounds when designing Central domain inhibitors.

- in the Introduction (lines 40-42), the phrase "In bacteria, they encode a unique Topo II called gyrase, which is a heterotetramer and has ability to introduce negative supercoils into closed-circular DNA." can be misinterpreted and a non-expert may think that bacteria only encode DNA gyrase, which is incorrect. This sentence should be clarified.

Our response: We thank the suggestion from reviewer. We have rephrased this part in our introduction.

- lines 314-315, the sentence "The results of AFM [...] bind the crossover site." needs improvement.

Our response: We appreciate the reviewer's suggestions and made improvement to this part in our revised manuscript.

Reviewer #3 (Remarks to the Author):

The provided additional information is still limited in situating the topoisomerase II of viruses in the subfamilies of Type II topoisomerases. Besides the presence in bacteria, eukaryotic cells, and viruses (as mentioned), Topo IIs can also be present in archaea, plants, and certain algae. In what aspects do Topo IIA differ from Type IIB topoisomerases?

Our response: We thank the suggestion from reviewer. We have revised the introduction, included subtypes classification, and provided a brief summary of their similarities and differences. Due to space constraints, further details should be referred to in the references.

Besides visualization, AFM can be used to obtain mechanistic insights and this should be improved.

In the provided pictures (Fig. 6), it can be observed that pP1192R can – besides binding to a cross-over - also bind to the linear DNA parts of the supercoiled plasmid.

Fig. 6b is not that clear. Fig. 6c is fine to better see that there is binding (the tilting angle should be increased, and the Z range reduced). Can this be done for all images in Fig. 6b?

Our response: We thank the reviewer for suggesting this. The lack of clarity in Fig. 6b can be attributed to the compression applied to the PDF during the submission process. The following images are our uncompressed Fig. 6b.

We could convert all the images in Fig. 6b into the format shown in Fig. 6c, but due to space limitations, we can only select one representative example to be displayed in Fig. 6c. We will upload the other Fig. 6b image converted into Fig. 6c format as the source data for interested readers.

In Fig. 6c, we have included the tilting angle. However, increasing the angle would affect the clarity of the complete view of the plasmid. Furthermore, we have reduced the Z value from 3.5 to 3.3, as suggested by the reviewer.

Also, it should be clearly shown how many pP1192R bind to the plasmid (see e.g. 10.1016/j.febslet.2011.08.051). A more detailed discussion should be added to the text (compare the results with other studies).

Our response: We have addressed the results concerning the number of pP1192R bound to one plasmid in supplementary Fig. 6d and compared it with the reference in the discussion section. We have included the total number of pP1192R bound to the plasmid in Fig. 6d.

Fig. 6d, left panel: Show the additional AFM images as Supplementary Material that support these

results.

Our response: There are a large number of additional AFM images, so we will upload them to the source data. The supplementary information does not have enough space to include all of them. However, we have selected some representative images here for your reference.

Could a bending of the DNA be observed for the duplex case? If so, make a graph “Frequency versus angle (0-180°)”. If not, show a few cases (3D image zoomed in) showing no bending. This figure should be discussed in the manuscript. Fig. 6d, right panel: This figure should be discussed in the manuscript. Also, the results on the relaxed plasmids show that pP1192R can bind to linear DNA. This can be added as Supplementary Material.

Our response: Yes, we observed the bending of DNA in the duplex case as shown in the picture (below) . However, the primary aim of using AFM was to verify if the protein has a preference for binding DNA crossovers. In the result section “In vitro interaction of pP1192R with DNA crossovers” we are not concerned about how pP1192R binds to DNA duplexes, and we have provided exact bending angle in our cryo-EM structures. Since these results (pP1192R binding to DNA duplex in AFM) are nearly of no relevance to this study, we have decided not to include them in the discussion section or supplementary material. And we have discussed Fig. 6d in the manuscript.

Supplementary Fig. 6a: Give also cross sections at a few locations supporting the measured average height.

Supplementary Fig. 6b: It seems that aggregation is present. Give a few cross sections for the non-aggregated protein. Also give some cross-section when the Four-way junction DNA protein is bound to the DNA (cross-over, linear) supporting the measured height of DNA and protein bound.

Our response: We thank the suggestions from the reviewer. We have included cross-section images in Supplementary Fig. 6a, b, and c. Additionally, we have added an image in Supplementary Fig. 6c showing pP1192R bound to Four-way DNA (two linear DNA duplex captured by pP1192R).

Supplementary Fig. 6c: Remove the “0” bar chart and give a graph “probability versus the number of Topo II on the plasmid” (see e.g. 10.1016/j.febslet.2011.08.051). Discuss these graphs in the manuscript.

Our response: we have removed the “0” bar chart and give a graph “probability versus the number of Topo II on the plasmid”. We have also discussed this graph in the discussion section of our study.

Image the four-way junction DNA complex: The shown image demonstrates that the sample has to be diluted. Nevertheless, larger four-way junctions could have been made (see e.g. 10.1016/j.febslet.2011.08.051).

Our response: The “Four-way junction DNA” in our manuscript refers to a specific type of DNA described in the reference (DOI:10.1093/nar/27.4.984.). This DNA is synthesized using four oligonucleotides, as illustrated in the figure (below).

[Redacted]

"Please refer to Figure 2A in West KL, Austin CA. Human DNA topoisomerase IIbeta binds and cleaves four-way junction DNA in vitro. *Nucleic Acids Res.* 1999 Feb 15;27(4):984-92. doi: 10.1093/nar/27.4.984. PMID: 9927730; PMCID: PMC148277."

In reference (DOI:10.1016/j.febslet.2011.08.051), the authors described that there is a cross-over instead of a four-way junction. They observed that Topo II captures two linear DNA duplexes, resulting in the formation of a cross-over structure (resembling the four-way junction DNA) . We also observed it in the linear plasmid images and included it in supplementary Fig. 6c.

REVIEWER COMMENTS

Reviewer #1 (Remarks to the Author):

The authors have addressed all of my doubts and concerns.

REVIEWER COMMENTS to the 2nd revised manuscript

Reviewer #3:

Besides visualization, AFM can be used to obtain mechanistic insights and this should be improved. In the provided pictures (Fig. 6), it can be observed that pP1192R can – besides binding to a cross-over - also bind to the linear DNA parts of the supercoiled plasmid.

Fig. 6b is not that clear. Fig. 6c is fine to better see that there is binding (the tilting angle should be increased, and the Z range reduced). Can this be done for all images in Fig. 6b?

Our response: We thank the reviewer for suggesting this. The lack of clarity in Fig. 6b can be attributed to the compression applied to the PDF during the submission process. The following images are our uncompressed Fig. 6b.

We could convert all the images in Fig. 6b into the format shown in Fig. 6c, but due to space limitations, we can only select one representative example to be displayed in Fig. 6c. We will upload the other Fig. 6b image converted into Fig. 6c format as the source data for interested readers.

- ⇒ The images can be cropped (no need to show the mica substrate around the DNA. In this case, multiple images could be shown.

In Fig. 6c, we have included the tilting angle. However, increasing the angle would affect the clarity of the complete view of the plasmid. Furthermore, we have reduced the Z value from 3.5 to 3.3, as suggested by the reviewer.

- ⇒ In the shown images in Fig. 6b, it is not clear that pP1192R is bound at a crossover site. It seems to be bound to a linear strand; in Fig. 6b 1, 2, 4, 5, 6, 7 and 9 (bottom pP1192R) pP1192R seems not to be bound at the crossover.
- ⇒ The additional provided AFM images do not show high resolution: it seems that P1192R is aggregated and individual P1192R cannot be distinguished.

Also, it should be clearly shown how many pP1192R bind to the plasmid (see e.g. 10.1016/j.febslet.2011.08.051). A more detailed discussion should be added to the text (compare the results with other studies).

Our response: We have addressed the results concerning the number of pP1192R bound to one plasmid in supplementary Fig. 6d and compared it with the reference in the discussion section. We have included the total number of pP1192R bound to the plasmid in Fig. 6d.

The results of AFM showed that the viral Topo II also prefers to bind the crossover site, particularly at larger crossing angles ($> 50^\circ$). Compared to the results obtained

with human Topo II²⁸, we found that the larger plasmid binds more Topo II in one plasmid.

- ⇒ The crossing angle drawn in Figure 6b 9 is not correct.
- ⇒ ... not clear what is meant by “particularly at larger crossing angles (>50°)”; how do these findings compare to the ones reported in ref. 28? The second sentence in the discussion is not clear.

Fig. 6d, left panel: Show the additional AFM images as Supplementary Material that support these results.

Our response: There are a large number of additional AFM images, so we will upload them to the source data. The supplementary information does not have enough space to include all of them. However, we have selected some representative images here for your reference.

- ⇒ The additional provided images have been scanned with a too-large scan size. Consequently, the exact binding location of the protein cannot be accurately enough observed.

Could a bending of the DNA be observed for the duplex case? If so, make a graph “Frequency versus angle (0-180°)”. If not, show a few cases (3D image zoomed in) showing no bending. This figure should be discussed in the manuscript. Fig. 6d, right panel: This figure should be discussed in the manuscript. Also, the results on the relaxed plasmids show that pP1192R can bind to linear DNA. This can be added as Supplementary Material.

Our response: Yes, we observed the bending of DNA in the duplex case as shown in the picture (below). However, the primary aim of using AFM was to verify if the protein has a preference for binding DNA crossovers. In the result section “In vitro interaction of pP1192R with DNA crossovers” we are not concerned about how pP1192R binds to DNA duplexes, and we have provided exact bending angle in our cryo-EM structures. Since these results (pP1192R binding to DNA duplex in AFM) are nearly of no relevance to this study, we have decided not to include them in the discussion section or supplementary material. And we have discussed Fig. 6d in the manuscript.

- ⇒ The provided AFM images do not show that the protein prefers binding at DNA crossovers (see comments before).
- ⇒ The added discussion related to Fig. 6d is not clear (see comments before).

Supplementary Fig. 6a: Give also cross sections at a few locations supporting the measured average height.

Supplementary Fig. 6b: It seems that aggregation is present. Give a few cross-sections for the non-aggregated protein. Also give some cross-section when the Four-way junction DNA protein is bound to the DNA (cross-over, linear) supporting the measured height of DNA and protein bound.

Our response: We thank the suggestions from the reviewer. We have included cross-section images in Supplementary Fig. 6a, b, and c. Additionally, we have added an image in Supplementary Fig. 6c showing pP1192R bound to Four-way DNA (two linear DNA duplex captured by pP1192R).

- ⇒ The height of the protein (cross-section Suppl. Fig. 6b) is much larger (~5 nm) than when it is bound to the DNA (~2.5 nm) (cross-section Suppl. Fig. 6c) due to the aggregation of the protein in Suppl. Fig. 6b. The adsorption buffer should be optimized. Aggregation can also be observed in Fig. 6b where the binding of the protein to the DNA is shown. This aggregation can influence the binding to the DNA.

Supplementary Fig. 6c: Remove the “0” bar chart and give a graph “probability versus the number of Topo II on the plasmid” (see e.g. 10.1016/j.febslet.2011.08.051). Discuss these graphs in the manuscript.

Our response: we have removed the “0” bar chart and give a graph “probability versus the number of Topo II on the plasmid”. We have also discussed this graph in the discussion section of our study.

- ⇒ As commented above, this sentence is not clear.

Image the four-way junction DNA complex: The shown image demonstrates that the sample has to be diluted. Nevertheless, larger four-way junctions could have been made (see e.g. 10.1016/j.febslet.2011.08.051).

Our response: The “Four-way junction DNA” in our manuscript refers to a specific type of DNA described in the reference (DOI:10.1093/nar/27.4.984.). This DNA is synthesized using four oligonucleotides, as illustrated in the figure (below).

In reference (DOI:10.1016/j.febslet.2011.08.051), the authors described that there is a cross-over instead of a four-way junction. They observed that Topo II captures two linear DNA duplexes, resulting in the formation of a cross-over structure (resembling the four-way junction DNA). We also observed it in the linear plasmid images and included it in supplementary Fig. 6c.

- ⇒ In Fig. 6c, it is not clear that this is a four-way junction or a resembling structure; it seems to be 2 pieces of DNA strands that are close to each other and the protein is bound to a linear strand.

REVIEWER COMMENTS

Reviewer #3 (Remarks to the Author):

We thank the suggestion from reviewer. However, we would like to emphasize that the focus of our manuscript is primarily a structural (solve the atomic structure of pP1192R) and functional (illustrate the duplex cleavage model of topo II) study. The purpose of involving atomic force microscopy in our study was simply to validate the propensity of our protein to bind to DNA crossover. We observed that the majority of protein binding to the plasmid in crossover rather than duplex positions provides sufficient evidence for our viewpoint. The AFM images we provided are in highest resolution we can obtain using our equipment, which are enough to support our conclusion (pP1192R binds to crossover sites). The specifics of how the protein binds to the crossover sites, as well as the exact binding mode, are not the focus of our manuscript and are not easily to be fulfilled in the short term. Indeed these specific questions are quite interesting and should be further explored in future.

The images can be cropped (no need to show the mica substrate around the DNA. In this case, multiple images could be shown.

In the shown images in Fig. 6b, it is not clear that pP1192R is bound at a crossover site. It seems to be bound to a linear strand; in Fig. 6b 1, 2, 4, 5, 6, 7 and 9 (bottom pP1192R) pP1192R seems not to be bound at the crossover. The additional provided AFM images do not show high resolution: it seems that P1192R is aggregated and individual P1192R cannot be distinguished.

Our response: We have converted all the original images in Fig. 6b into the 3D format shown as below, and also, we have uploaded them into Figshare.com (DOI: 10.6084/m9.figshare.25273231), which can be downloaded.

We must emphasize that, in the original Fig 6b, the protein shown are binding to the crossover regions. As you see the schematic diagram below, superhelical plasmids are highly flexible in solution, with quite different conformations and degrees of superhelix (also see supplementary Fig. 6a). Some have a low degree of superhelix, which looks like "8"; but in some cases, they have a high degree of superhelix, just appearing like a "pseudo-linear" DNA, but they are thicker than duplex DNA, which are easily distinguished in provided images. Therefore, we have every reason to think all those proteins are bound to crossover sites.

However, to avoid any misunderstandings, we have updated new images which provide clearer evidence of proteins bound to crossover sites in fig 6b. As suggested, we have also provided all 3D images in fig 6c.

The AFM images we provided are in highest resolution we can obtain using our equipment (Dimension FastScan Bio (Bruker)), which are already enough to support our conclusion (pP1192R binds to crossover sites). As for protein aggregation, the affinity purified proteins were further purified through SEC, ensuring the great majority were homogeneous. Based on electron microscopy structure, the structure of pP1192R is not a sphere. Therefore, different orientations of protein on mica surface can lead to variations in height and size measurements. Besides, this kind of protein itself may form polymeric complexes in solution, as described in DOI: 10.7554/eLife.41215 (figure below). On the other hand, it is not certain whether a crossover site

can bind only one protein molecule, which is one of the undetermined questions in this field. But in any case, it doesn't affect the fact that the proteins are bound to the crossover sites of supercoiled plasmids.

[Redacted]

"Please refer to Figure 2A in Soczek KM, Grant T, Rosenthal PB, Mondragón A. CryoEM structures of open dimers of gyrase A in complex with DNA illuminate mechanism of strand passage. *Elife*. 2018 Nov 20;7:e41215. doi: 10.7554/eLife.41215. PMID: 30457554; PMCID: PMC6286129."

The crossing angle drawn in Figure 6b 9 is not correct.

Our response: We thank reviewer pointing this. The crossing angle in ref.28 is measured in linear sample (two DNA duplex are simultaneously captured by topo II). The angle we measured is in plasmid sample. Due to the influence of the plasmid's intrinsic helical and flexible structure, the crossing angle is difficult to define and measure accurately, however, it would not affect our conclusion.

In our linear DNA samples, we can hardly find pP1192R binding to the linear DNA, not to mention "four-way" crossover, therefore there is no statistical significance measuring crossing angles in these samples. The viewpoint of our manuscript is that pP1192R binds to crossovers, and the angles are unrelated to this point. Thus, we have deleted this section.

not clear what is meant by "particularly at larger crossing angles ($>50^\circ$)"; how do these findings compare to the ones reported in ref. 28? The second sentence in the discussion is not clear.

Our response: we have deleted the description about crossing angle and rephased the discussion "The results of AFM showed that the viral Topo II also prefer to bind the crossover site in a way similar to human Topo II²⁸.". The crossing angle doesn't matter much to our main point of view.

The additional provided images have been scanned with a too-large scan size. Consequently, the exact binding location of the protein cannot be accurately enough observed.

Our response: The scan size we scanned are the same as ref.7, and they are clear enough to demonstrate the exact binding locations as you can see below. Both 3D-image and cross section

image have clearly shown the exact binding of pP1192R to crossover sites.

The provided AFM images do not show that the protein prefers binding at DNA crossovers (see comments before).

The added discussion related to Fig. 6d is not clear (see comments before).

Our response:

In Fig. 6b, indeed, some proteins are bound to highly supercoiled regions, creating a

misunderstanding for the reviewers that these regions appear linear. However, they are actually regions where two DNA duplexes are tightly intertwined, as explained before. To avoid any misunderstandings, we have updated images that provide clearer evidence of crossovers bound with pP1192R.

Also we have deleted the crossing angle section (Fig. 6d) and rephased our discussion.

The height of the protein (cross-section Suppl. Fig. 6b) is much larger (~5 nm) than when it is bound to the DNA (~2.5 nm) (cross-section Suppl. Fig. 6c) due to the aggregation of the protein in Suppl. Fig. 6b. The adsorption buffer should be optimized. Aggregation can also be observed in Fig. 6b where the binding of the protein to the DNA is shown. This aggregation can influence the binding to the DNA.

Our response: As we mentioned before, in our linear DNA samples, we can hardly find pP1192R binding to the linear DNA, not to mention the “four-way” crossover. We have repeated experiments, but obtained no better results than the original one (original Suppl. Fig. 6c). Also, we must address that ~2.5nm (exactly 2.7nm) is enough to distinguish between protein and DNA duplex, whose height is less than 1.5nm. To avoid misunderstanding, we have updated a new AFM image of pP1192R bound to the crossover site along with its cross-section image.

As for aggregation, the affinity purified proteins were further purified through SEC, ensuring the great majority were homogeneous (All buffers have been optimized). Based on electron microscopy structure, the actual structure of 1192 is not a sphere. Therefore, different orientations of protein on mica surface can lead to variations in height measurements (2.5nm-5.0nm indicates individual protein particle). Besides, this kind of protein itself may form polymeric complexes in solution, referring to DOI: 10.7554/eLife.41215 (In fact, we indeed observed polymeric complexes in 2D classification of cryo-EM data processing). On the other hand, it is not certain whether a crossover site can bind only one protein molecule, which is one of the undetermined questions in this field. But in any case, it doesn't affect the fact that the proteins are bound to the crossover sites of supercoiled plasmids.

As commented above, this sentence is not clear.

Our response: We have rephrased this sentence in discussion.

In Fig. 6c, it is not clear that this is a four-way junction or a resembling structure; it seems to be 2 pieces of DNA strands that are close to each other and the protein is bound to a linear strand.

Our response: As mentioned before, we have redone the assay many times, but obtained nothing better than the original one (original Suppl. Fig. 6c). It's indeed difficult to distinguish between the four-way junction or the resembling structure as suggested. And it doesn't affect our point of view in the manuscript. Therefore, to avoid misunderstanding, we have exchanged the original one with a new image of pP1192R bound to crossover site.